# Epidemiological characteristics and transmission dynamics of dengue fever in China

Haobo Ni[1,8], Xiaoyan Cai [1,8], Jiarong Ren [2,8], Tingting Dai [1], Jiayi Zhou [1], Jiumin Lin [3], Li Wang [4], Lingxi Wang [1], Sen Pei [5], Yunchong Yao [1], Ting Xu [1], Lina Xiao[1], Qiyong Liu [2,6,7] ✉, Xiaobo Liu [2,6,7] ✉ & Pi Guo [1] ✉

China has experienced successive waves of dengue epidemics over the past decade. Nationwide data on 95,339 dengue cases, 89 surveillance sites for mosquito density and population mobility between 337 cities during 2013-20 were extracted. Weekly dengue time series including time trends and harmonic terms were fitted using seasonal regression models, and the amplitude and peak timing of the annual and semiannual cycles were estimated. A data-driven model-inference approach was used to simulate the epidemic at city-scale and estimate time-evolving epidemiological parameters. We found that the geographical distribution of dengue cases was expanding, and the main imported areas as well as external sources of imported cases changed. Dengue cases were predominantly concentrated in southern China and it exhibited an annual peak of activity, typically peaking in September. The annual amplitude of dengue epidemic varied with latitude ($F = 19.62$, $P = 0.0001$), mainly characterizing by large in southern cities and small in northern cities. The effective reproduction number $R_{eff}$ across cities is commonly greater than 1 in several specific months from July to November, further confirming the seasonal fluctuations and spatial heterogeneity of dengue epidemics. The results of this national study help to better informing interventions for future dengue epidemics in China.

As one of the key neglected tropical diseases (NTDs)[1,2], dengue fever is a viral infection transmitted to humans through the bite of infected mosquitoes, primarily Aedes albopictus and Aedes aegypti[3]. The global incidence of dengue fever has notably increased over the past two decades. According to the World Health Organization (WHO), reported cases surged tenfold from 2000 to 2019, escalating from 0.5 million to 5.2 million cases[4]. Since 2023, sustained transmission of dengue, coupled with an unexpected upsurge in cases, has led to over 5 million infections reported across more than 80 countries/territories, which included more than 5000 dengue-related deaths[4,5]. Therefore, dengue fever poses a significant threat to the health of billions of individuals and has a substantial burden worldwide.

[1]Department of Preventive Medicine, Shantou University Medical College, Shantou, China. [2]National Key Laboratory of Intelligent Tracking and Forecasting for Infectious Diseases, National Institute for Communicable Disease Control and Prevention, Chinese Center for Disease Control and Prevention, Beijing, China. [3]Department of Hepatology and Infectious Diseases, Second Affiliated Hospital of Shantou University Medical College, Shantou, China. [4]Department of Medical Statistics and Epidemiology, School of Public Health, Sun Yat-sen University, Guangzhou, China. [5]Department of Environmental Health Sciences, Mailman School of Public Health, Columbia University, New York, New York, USA. [6]Department of Vector Control, School of Public Health, Cheeloo College of Medicine, Shandong University, Jinan, China. [7]Xinjiang Key Laboratory of Vector-borne Infectious Diseases, Urumqi, Xinjiang, China. [8]These authors contributed equally: Haobo Ni, Xiaoyan Cai, Jiarong Ren. ✉e-mail: liuqiyong@icdc.cn; liuxiaobo@icdc.cn; pguo@stu.edu.cn

Approximately 75% of the global prevalence for dengue infections locates in Southeast Asia and the Western Pacific regions close to the tropics[6–8]. In China, the expansion of dengue poses a significant public health concern. Prior to the year 2000, occurrences of dengue fever in China were predominantly localized in tropical regions along the southern coastline[9]. However, there has been a gradual northward progression of dengue outbreaks over time[10]. An epidemic of dengue fever occurred in China during 2014, with the number of infected people reaching more than 40,000[11,12]. In addition, our previous study suggests that frequent human travel increases the possibility that individuals will be exposed to disease vectors, allowing the rapid spread of dengue virus (DENV) to susceptible populations[13]. Some previous studies have identified long-term urbanization, human mobility and population growth are main influential drivers shaping dengue's distribution and dynamics[14,15]. In addition, the interactions between recent climate change and key socio-environmental drivers have also been shown to contribute to dengue's expansion[16]. Although our previous studies have concerned about the national burden of dengue fever, the studies mainly focused on identifying the connectivity between local and imported cases as well as the fundamental factors for dengue epidemics[17,18]. In addition, a recent study has projected future risk of dengue related to hydrometeorological conditions in China and quantify its associated disease burden in a future with a changed climate[19]. However, we have limited knowledge on the time-evolving transmission dynamic features and burden associated with nationwide spread of dengue fever in China up to now.

To understand the transmission of the DENV and better control its progression in the future, it is critical that the epidemiological characteristics underpinning these outbreaks are quantified and analyzed in space and time. Here, we performed this large-scale study to characterize the epidemiology of dengue fever based on national surveillances. A city-resolved metapopulation network model was used to simulate the transmission of dengue infection among the 337 cities in China. The metapopulation network model can flexibly generate patterns of complex population movements between different regions. The model depicts national data on dengue infections and is coupled with an iterative Bayesian inference algorithm, named the ensemble adjustment Kalman filter (EAKF), which assimilates observations of daily cases in each city, as well as human movement between cities[12,13]. The incorporation of Bayesian inference supports a fitting of the model to case observations and estimation of the unobserved state variables (for example, new infections per day within a city) and system parameters (for example, transmission rate in each city). Our findings provide the most comprehensive quantification to date of the time-evolving epidemiological features and burden associated with successive epidemics waves of dengue fever in China over the period of 2013–2020.

## Results
### Seasonal and pandemic characteristics of dengue fever
From 2013 to 2020, a total of 95,339 cases of dengue fever were reported in investigated 337 cities in China, including 83,318 local cases and 12,021 imported cases. In terms of spatial distribution, dengue fever cases were mainly distributed in the southeastern part of China, specifically, south of the Hu line (Fig. 1a). Over the span of eight years, there was a noticeable expansion in the geographical spread of dengue cases, with a particular increase observed in imported cases (Figure S2, S3). Almost all cities in south of the Hu line were reported to have imported dengue infections from outside the country in 2019 (Figure S3). In terms of temporal distribution, dengue fever in China usually has an epidemic peak every year, usually between July and November, with sporadic cases at other times (Fig. 1b, c, g). Among 83,318 local dengue cases, 47,100 (56.53%) of the reported dengue cases were laboratory-confirmed cases, 34,986 (41.99%) were clinically diagnosed cases, and 1232 (1.48%) were probable cases. Typically, the

diagnosis of reported dengue cases was predominantly laboratory-confirmed. However, during the stronger dengue outbreaks in 2014 and 2019, more than half of the cases were clinically diagnosed (Fig. 1d).

From 2013 to 2020, Myanmar and Cambodia emerged as the primary original countries with imported dengue cases, with most of imported cases being brought into the Yunnan and Guangdong provinces in China. Other provinces, except Xinjiang and Tibet, also reported dengue fever cases imported from abroad (Fig. 1e, f). The proportion of imported dengue cases from original countries varied in different years. Myanmar emerged as the primary importing country during the period of 2013–2017, while Vietnam replaced this role during 2018–2020 (Figures S4, S5). Dengue cases were mainly imported into Yunnan province in 2013–2017 and to Guangdong province in 2018–2020 (Figures S4, S5). The temporal distribution of imported cases exhibited an upward trend starting in March, reaching a peak in October, followed by a decline (Fig. 1g).

Except for the low incidence of dengue fever among children and adolescents aged 0–19 years, there was no substantial difference in the incidence of dengue fever among local dengue cases across different age groups (Figure S6). The incidence rate was slightly higher in young adults aged 20–49 years in Guangxi than in other age groups, and slightly higher in the individuals over 60 years in Fujian than in other age groups.

The median time from onset to diagnosis was consistent at 4 days (IQR 2–7 days) for both local dengue cases and imported cases. The probability density distribution from onset to diagnosis showed that the time to diagnosis was slightly shorter for reports in the months following the start of the annual dengue outbreak than before the outbreak (Figure S7). The probability density distributions of incidence to diagnosis were exhibited similar for different age groups (Figure S7).

Nationwide, local cases of dengue fever exhibited an annual peak of activity, typically peaking in September. There was no significant difference ($P = 0.3514$) in the time of reaching the annual peak of dengue epidemic across difference provinces (Fig. 2c, f). With increasing dimensions of dengue epidemics, the annual amplitude fluctuations decreased ($P = 0.0001$) (Fig. 2a, d). Outside the annual amplitude of dengue, there were more sporadic cases in the south than in the north, with consistent of results for semi-annual periodicity ($P = 0.0016$) (Fig. 2b, e).

The annual amplitude trend of dengue imported cases is consistent with local cases, showing a trend of strong in the South and weak in the north ($P = 0.0179$), with no significant difference in the semi-annual periodicity between the north and south regions ($P = 0.8988$) (Figure S19). The peak of imported dengue cases reached their annual peak later in the southern region than in the northern region, which was statistically significant ($P = 0.0015$) (Figure S15-S19).

### Intrinsic dynamic characteristics of DENV transmission
By fitting the inferred model of DENV transmission to the daily observed case, we successfully captured dengue outbreaks at city, urban agglomerations, and national scales in 2019 (Fig. 3). The peak times of dengue outbreaks were very consistent across regions, but there were still some inconsistencies. For example, in the Yunnan Guangxi border area, the infections number peaked in mid-September, while in the Pearl River Delta, infections peaked in mid-October (Fig. 3c, d). We also estimated the number of monthly total infections (reported and unreported infections) across the country in 2019. Most dengue infections were unreported cases (Fig. 3b). It was estimated that the total number of dengue infections in the country peaks in September, with 25,691 (95% CI 23,214-28,333) new infections in one month (Fig. 3b).

Next, we focused on these major urban agglomerations and the city scales in 2019. Overall, the estimated number of infections and

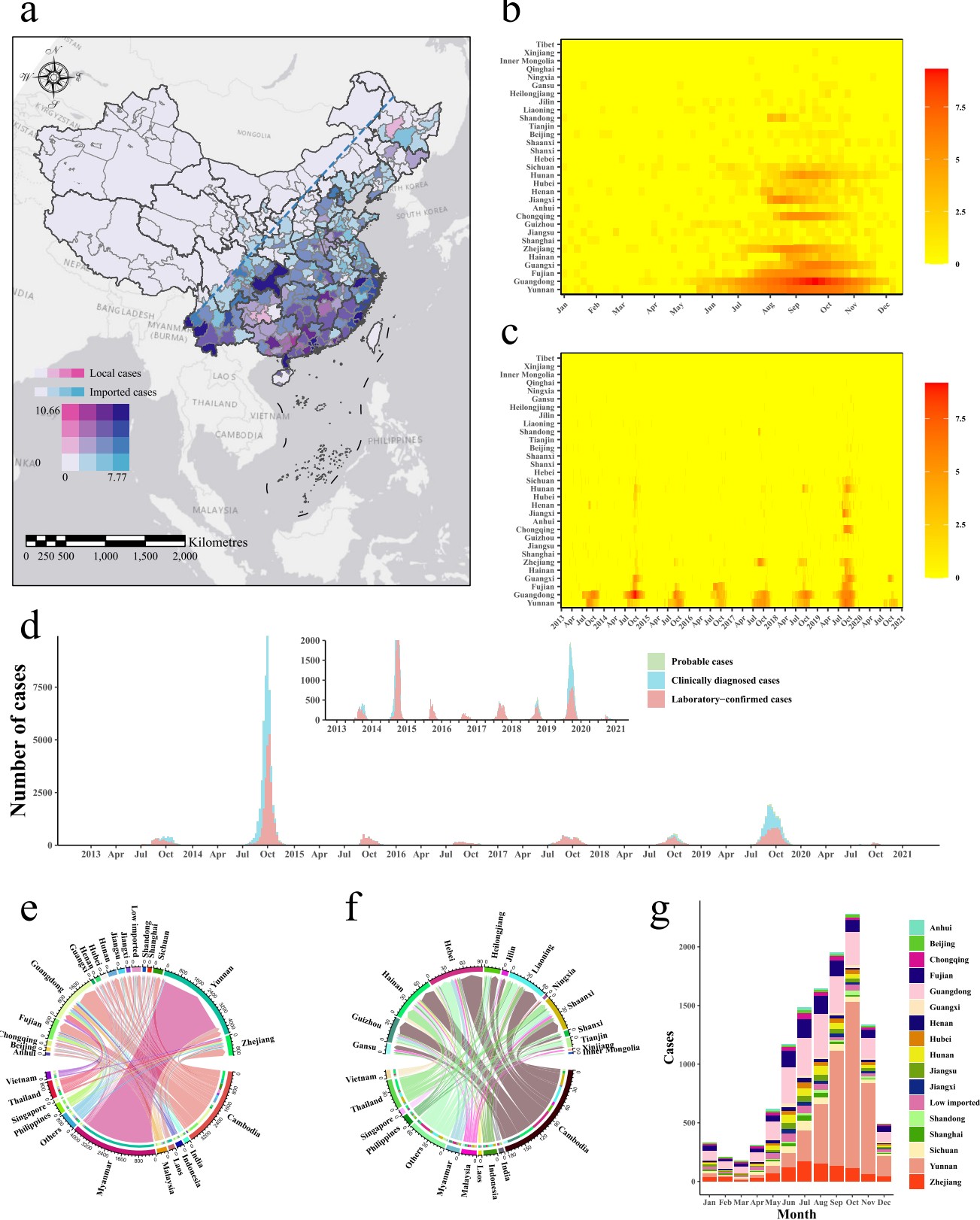

related parameters across the regions began to rise in June, peaked around September and October, and then declined sharply (Fig. 4). The Yunnan Guangxi border region, the Fujian Zhejiang coast area, and Chongqing city had the highest estimated number of infections in September, with 11,659 (95% CI 10,025-14,019), 3063 (95% CI 2619–3524), and 2527 (95% CI 2141–2971) cases, respectively (Fig. 4a, c,

d). The highest estimated number of infections in the Pearl River Delta occurred in October, with 2674 (95% CI 2296–3053) cases (Fig. 4b). Changes in transmission rates were broadly consistent across the four agglomerations regions (Fig. 4e–h). The $R_{eff}$ began to rise gradually in June, exceeding one during the dengue epidemic, and then began to fall in October until it tended to zero (Fig. 4i–l). The force of dengue

**Fig. 1 | Spatial and temporal distribution of local and imported cases of dengue fever, 2013-2020. a** Spatial distribution of local and imported dengue cases (log-transformed scale). The shades of red in the city blocks indicate the severity of local cases, with three breakpoints: 1.10, 3.22 and 5.88, while the shades of blue in the city blocks indicate the severity of imported cases, with three breakpoints: 0.69, 2.40 and 4.11. The Jenks natural breaking method was used for data classification. The blue dashed line is Hu Line, a geographical demarcation line for population distribution and economic development in China. **b** Seasonal distribution of dengue cases, plotted as the sum of the weekly number of cases throughout the year from 2013 to 2020. **c** Time series of weekly cases of dengue. For (**a**)–(**c**), natural logarithmic transformation (ln(cases + 1)) is applied. As a widely used data

normalization method, logarithm transformation of data was performed to make the data distribution more uniform in order to better explore the seasonal fluctuation characteristics of dengue incidence. **d** Time series of weekly probable, clinically diagnosed and laboratory-confirmed cases of dengue. The subfigure providing a zoomed-in view of the weekly dengue cases on the vertical coordinate. **e** Source-sink relationships between original countries with imported dengue cases and Chinese provincial administrations with relatively high imported cases. **f** Relationships between original countries with imported dengue cases and Chinese provinces with relatively low imported cases. **g** Temporal distribution of imported dengue cases across provinces.

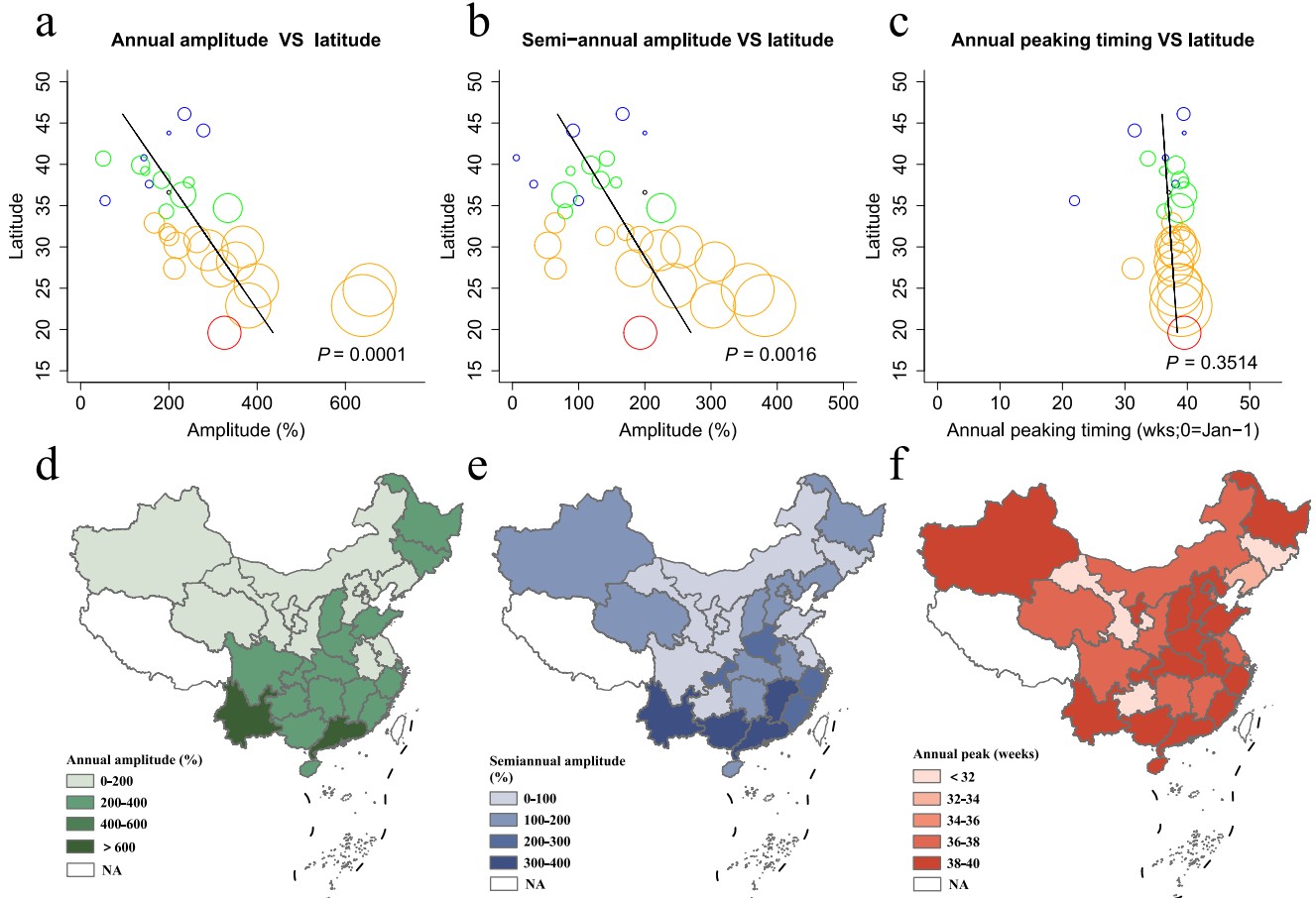

**Fig. 2 | Characteristics of a latitudinal gradient in the periodicity and peak timing of dengue fever epidemics in China. a** Amplitude of the annual periodicity ($F = 19.62$, $P = 0.0001$). **b** Amplitude of the semi-annual periodicity ($F = 12.24$, $P = 0.0016$). **c** Timing of primary annual dengue fever peak, in weeks from 1st January ($F = 0.90$, $P = 0.3514$). Colors represent different climatic zones (red = tropical, yellow = sub-tropical, green = warm-temperate, blue = mid-temperate, black = cold-temperate). Symbol size is proportional to the number of dengue cases in each province. The solid black line indicates a linear regression fit (regression weighted by the average annual number of dengue fever cases). The results of the $F$ test and

the corresponding $P$ values were employed to testing the significance of the linear regression fit. $P$ values are given on the graphs. **d** Amplitude of the annual periodicity in each province. **e** Amplitude of the semiannual periodicity in each province. **f** Timing for primary annual peak of dengue fever in each province. NA in the legend indicates that the region is temporarily unable to calculate these indicators (No dengue cases are reported in Tibet during the study time, and data for Hong Kong, Macau, Taiwan, Spratly Islands, and other parts of China outside of mainland China are not currently available).

infection was low in the four regions, with only a small increase during the epidemic (Fig. 4m–p). We also estimated the number of infections in four major urban agglomerations and all 337 cities, as well as parameter inferred information (Table 1, Table S4).

The variation of $R_{eff}$ was crucial for revealing the intrinsic transmission dynamics characteristics of DENV. We found that the $R_{eff}$ of dengue fever varied between cities across the nation (Fig. 5). In general, the $R_{eff}$ started to rise in June, reached a maximum around September and October, and then decreased sharply to

around zero. Cities with high $R_{eff}$ was concentrated in the south of China, especially some border and coastal cities. The analysis results of the effective reproduction number $R_{eff}$ across cities suggested that the current epidemic of dengue in China is not a long-term sustained epidemic state. The results of model inference further confirm the seasonal fluctuations of dengue epidemic, but also suggest that seasonal dengue transmission are spatially heterogeneous. We similarly gave the change in dengue transmission rates over time by city (Figure S21).

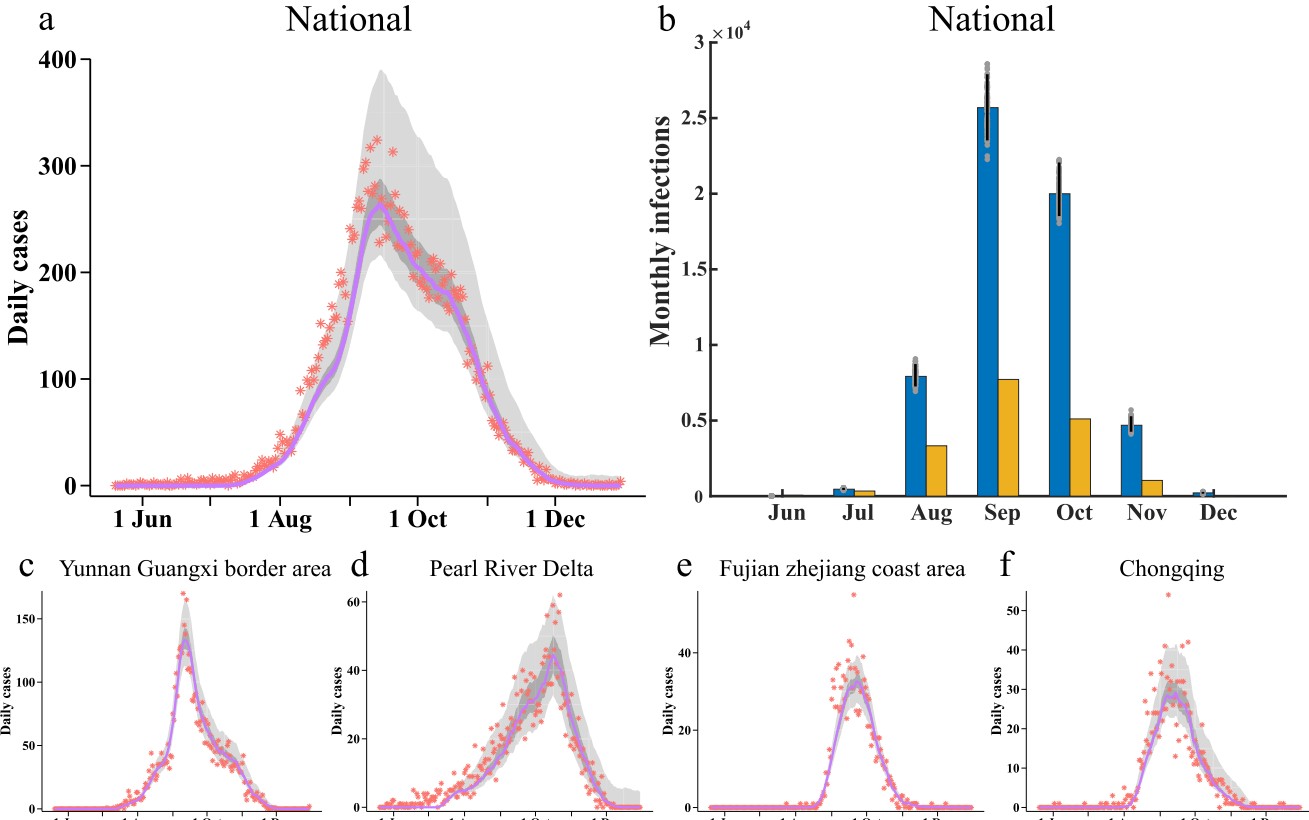

**Fig. 3 | Model fitting and estimation of national dengue infections. a, c–f** Model fit for the number of daily dengue cases (orange star symbols) in China as a whole and for four urban agglomerations. The solid purple line indicates the median of the estimates. Dark shaded intervals and light shaded intervals indicate the interquartile range and 95% confidence intervals (CIs) of the estimates, respectively. **b** Confirmed dengue cases (yellow bars) and estimated monthly total infections (blue bars) in China. Distributions were obtained from 300 ensemble members. The blue bars represent medians and whiskers show 95% CIs.

We revealed the infections in the main months of seasonal epidemics in each city, as well as the infections throughout the annual period. In August, infections were predominantly observed in coastal and border cities, whereas in September and October, dengue cases were reported in some inland cities located in the southern regions. (Fig. 6a–e). We further depicted the network of population movement and the relationship between the infections number resulting from population movement (Fig. 6f–j). Infections resulting from population movement were mainly found in the southern pivotal cities, which were centers of high population movement, and radiated to around cities. Additional information regarding the network of population movements and the number of infections resulting from population movements throughout the year and is given in the Supplementary Information file (Figure S22).

The model inference results are robust to the setting of parameters and variables (Figure S25-29). Furthermore, we generated simulated outbreaks based on different scenarios. Tests on fitting simulated dengue epidemics demonstrate that the inference method can recover key time-varying parameters in different simulation scenarios (Figure S24). We also did model fitting for dengue case data from 2013 to 2018 and estimated the total number of actual infections per month (Figure S30). The inferred results of this study demonstrated the inter-annual variations in epidemic size from 2013 to 2019, but it exhibited a consistent peak in annual activity over time (Fig. 3 and Figure S30).

## Discussion
Our study of more than 90,000 cases of dengue fever reported to the NNDSS during 2013–2020 in China gives a comprehensive quantification of the national burden of dengue disease and its intrinsic dynamics over time up to now. The geographical distribution of dengue cases was expanding in recent years, with obvious changes in the main imported areas as well as external sources of imported cases over the study period. The main external source country of imported cases changed from Myanmar to Cambodia and the main imported province changing from Yunnan to Guangdong. Our large study covers 337 climatologically diverse cities and suggests that although cases of dengue tend to arise in the autumn of the year throughout the country, annual amplitude of epidemics varies by latitude. The peak intensity mainly characterized by high in southern cities and low in northern cities. Our study demonstrated that the metapopulation network model coupled with an iterative Bayesian inference algorithm provided accurate simulations for city-level dengue transmission trajectories in retrospective forecasts of 337 cities in China. The proposed model accurately predicted the magnitude and temporal peak of seasonal epidemics as well as outbreaks of dengue.

Our study revealed that the age profile of dengue infection was in consistency with previous studies found in Asia[20,21]. Our study indicated a slightly higher incidence of dengue in the elderly in several regions, most likely due to a gradual decline in their immunity with age. A similar situation exists for young adults in Guangxi, which is located in southern China. Overall, there was little difference between the different age groups. The slight difference of dengue incidence among different age groups reminds us that the whole population may be susceptible to dengue infection.

A national, population-representative study of the spatio-temporal disease burden and characteristics of dengue infections could significantly enhance comprehension of the disease dynamics

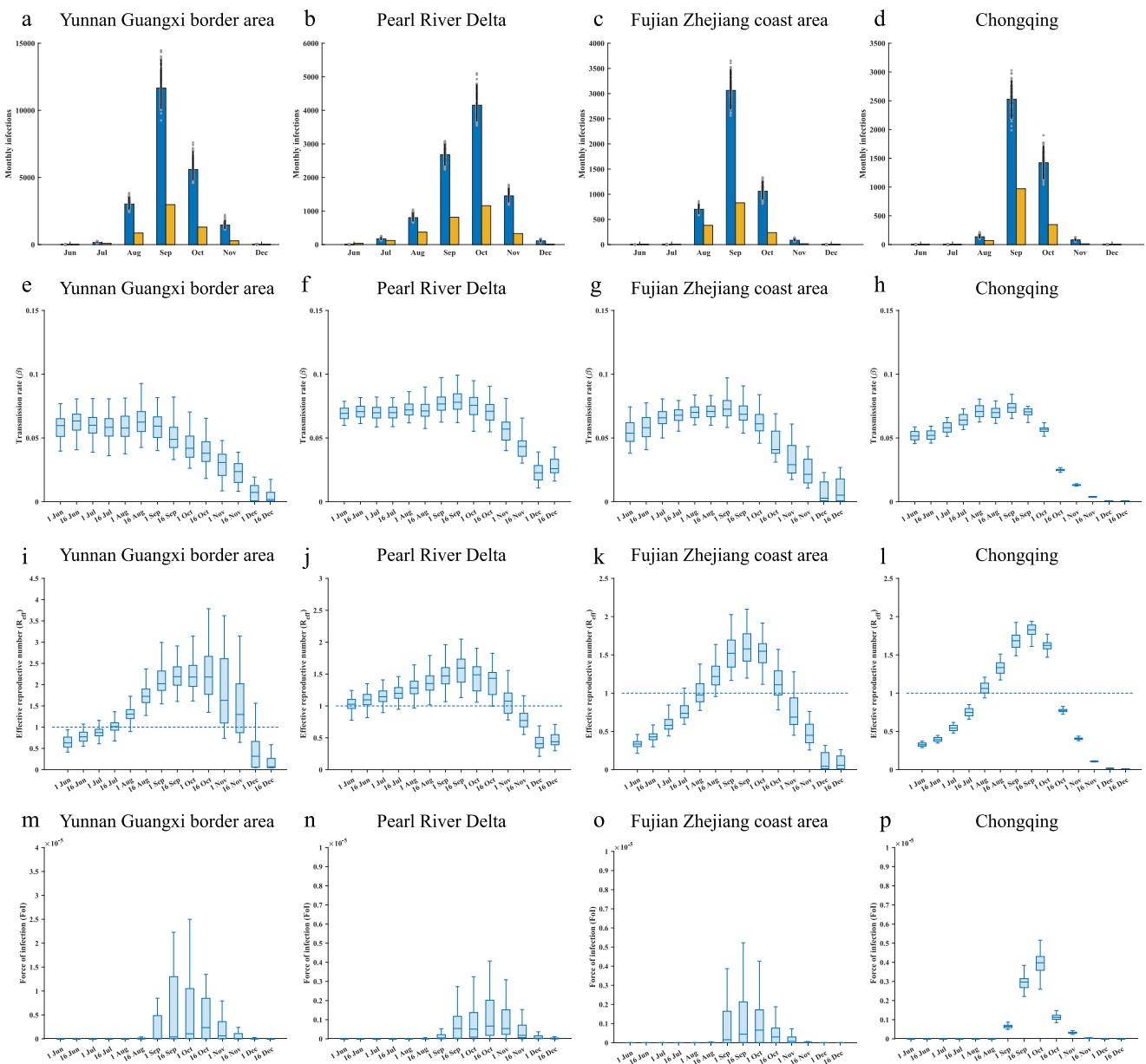

**Fig. 4 | Estimation of dengue infections and inference of time-varying transmission parameters in four urban metropolitans. a–d** Confirmed dengue cases (yellow bars) and estimated monthly total infections (blue bars) in four urban metropolitans. Distributions were obtained from 300 ensemble members. The blue bars represent medians and whiskers show 95% CIs. **e–h** Distribution of dengue transmission rates. **i–l** Distribution of dengue effective reproduction number. (**m–p**) Distribution of dengue force of infection. The center and box boundaries represent the median (50th), 75th, and 25th percentiles, respectively. In the box plots, the poles of the whiskers represent the distance from the 75th percentile to the maxima and the 25th percentile to the minima, respectively. The posterior estimate for the day is also displayed. The distribution is derived from 300 ensembled members.

over time scales as well as across geographical regions. By inferring the results of dengue infections in China from 2013–2019, we found that the seasonal trend of dengue epidemics was consistent, but the interannual intensity varied across years. This may due to the fact that local conditions (i.e., meteorological factors, mosquito density, urbanization growth, and other unknown factors) and population mobility jointly determine the epidemic size in most years. Results from similar studies also support the assumption that seasonal epidemics of dengue are accompanied by variations in epidemic size due to inter-annual variations in local conditions and other factors[22–24]. This study also revealed that dengue infection was predominantly occurred in the south-eastern region of China. Over the span of eight years, there was a noticeable expansion in the geographical spread of dengue cases,

especially the imported cases. One of the reason for these transmission trends might be the temperature changes due to climatic hazards, which expanded the area suitable for mosquito breeding[25,26]. Additionally, the uncontrolled urban growth, rapid and global traveling, difficulty of controlling vector spread, and environmental degradation were also potential factors contribute to the spread of dengue[18,27–30]. Further research is required to elucidate the underlying factors contributing to the dengue transmission.

Regarding the regional distribution of dengue cases, our analysis based on Bayesian inference algorithm coupled with disease transmission dynamics model revealed dengue outbreaks at the national, city, and major urban agglomerations scale. The results indicated that this acute highly infectious mosquito-borne infection is endemic in

**Table 1 | Parameters estimation derived from transmission model in four major urban agglomerations**

| Province | City | Estimated total infection | Transmission rate | Effective reproduction number | Force of infection |
|---|---|---|---|---|---|
| Zhejiang | Ningbo | 8 (4, 16) | 0.0547 (0.0493, 0.0604) | 1.4166 (1.2767, 1.5668) | $5.1 \times 10^{-10}$ ($2.8 \times 10^{-10}$, $2.3 \times 10^{-8}$) |
| | Wenzhou | 732 (571, 1004) | 0.0545 (0.0494, 0.0605) | 1.4646 (1.327, 1.6244) | $1.2 \times 10^{-6}$ ($7.6 \times 10^{-7}$, $2.0 \times 10^{-6}$) |
| | Zhoushan | 0 (0, 5) | 0.049 (0.0435, 0.055) | 1.1937 (1.0559, 1.344) | $8.8 \times 10^{-13}$ ($7.8 \times 10^{-13}$, $7.4 \times 10^{-8}$) |
| | Taizhou | 21 (13, 38) | 0.05 (0.0439, 0.0557) | 1.2673 (1.1119, 1.4155) | $3.2 \times 10^{-9}$ ($1.4 \times 10^{-10}$, $3.6 \times 10^{-8}$) |
| Fujian | Fuzhou | 1859 (1504, 2386) | 0.0658 (0.0606, 0.0723) | 1.5634 (1.4384, 1.7184) | $3.1 \times 10^{-6}$ ($2.1 \times 10^{-6}$, $4.7 \times 10^{-6}$) |
| | Xiamen | 92 (64, 141) | 0.0668 (0.0607, 0.076) | 1.4608 (1.3254, 1.6657) | $1.2 \times 10^{-7}$ ($5.0 \times 10^{-8}$, $3.1 \times 10^{-7}$) |
| | Putian | 1297 (1017, 1685) | 0.0683 (0.0615, 0.0749) | 1.5144 (1.3595, 1.6634) | $4.4 \times 10^{-6}$ ($2.7 \times 10^{-6}$, $6.9 \times 10^{-6}$) |
| | Quanzhou | 332 (254, 436) | 0.0537 (0.0501, 0.061) | 1.2133 (1.1293, 1.3795) | $4.5 \times 10^{-7}$ ($2.6 \times 10^{-7}$, $7.7 \times 10^{-7}$) |
| | Zhangzhou | 341 (266, 485) | 0.0645 (0.0592, 0.073) | 1.4019 (1.2833, 1.5872) | $6.4 \times 10^{-7}$ ($3.5 \times 10^{-7}$, $1.1 \times 10^{-6}$) |
| | Ningde | 193 (146, 279) | 0.0504 (0.0455, 0.0565) | 1.1199 (1.0087, 1.2567) | $7.0 \times 10^{-7}$ ($3.1 \times 10^{-7}$, $1.3 \times 10^{-6}$) |
| Guangdong | Guangzhou | 4755 (3901, 5918) | 0.0805 (0.0742, 0.0869) | 1.5803 (1.4568, 1.7082) | $2.3 \times 10^{-6}$ ($1.7 \times 10^{-6}$, $3.2 \times 10^{-6}$) |
| | Shenzhen | 878 (696, 1180) | 0.0735 (0.0659, 0.0822) | 1.5146 (1.3564, 1.6958) | $5.5 \times 10^{-7}$ ($3.5 \times 10^{-7}$, $8.7 \times 10^{-7}$) |
| | Zhuhai | 81 (63, 140) | 0.0777 (0.0698, 0.0881) | 1.5572 (1.3968, 1.7688) | $1.2 \times 10^{-7}$ ($3.2 \times 10^{-8}$, $4.5 \times 10^{-7}$) |
| | Foshan | 1232 (1037, 1494) | 0.0671 (0.0623, 0.0764) | 1.2326 (1.1428, 1.404) | $7.0 \times 10^{-7}$ ($4.7 \times 10^{-7}$, $1.0 \times 10^{-6}$) |
| | Jiangmen | 852 (670, 1053) | 0.078 (0.0707, 0.0864) | 1.5966 (1.4454, 1.7695) | $1.6 \times 10^{-6}$ ($9.7 \times 10^{-7}$, $2.5 \times 10^{-6}$) |
| | Zhaoqing | 60 (42, 96) | 0.0625 (0.0572, 0.0725) | 1.2958 (1.184, 1.5066) | $2.5 \times 10^{-8}$ ($5.1 \times 10^{-9}$, $1.3 \times 10^{-7}$) |
| | Huizhou | 215 (162, 274) | 0.0661 (0.0591, 0.0739) | 1.1991 (1.0706, 1.3426) | $1.1 \times 10^{-7}$ ($4.8 \times 10^{-8}$, $2.9 \times 10^{-7}$) |
| | Dongguan | 488 (377, 579) | 0.0683 (0.0621, 0.0773) | 1.1563 (1.0512, 1.3107) | $1.8 \times 10^{-6}$ ($9.3 \times 10^{-8}$, $3.3 \times 10^{-7}$) |
| | Zhongshan | 778 (605, 1006) | 0.0791 (0.0717, 0.0886) | 1.5136 (1.3704, 1.6978) | $1.4 \times 10^{-6}$ ($8.8 \times 10^{-7}$, $2.1 \times 10^{-6}$) |
| Guangxi | Fangchenggang | 471 (344, 671) | 0.0626 (0.0582, 0.0698) | 1.998 (1.8548, 2.235) | $6.6 \times 10^{-6}$ ($3.4 \times 10^{-6}$, $1.2 \times 10^{-5}$) |
| | Baise | 42 (24, 82) | 0.0509 (0.046, 0.0572) | 1.6587 (1.4964, 1.8639) | $1.7 \times 10^{-8}$ ($2.0 \times 10^{-10}$, $2.1 \times 10^{-7}$) |
| | Chongzuo | 62 (40, 106) | 0.0581 (0.0526, 0.066) | 1.8751 (1.6947, 2.1354) | $1.6 \times 10^{-7}$ ($2.7 \times 10^{-8}$, $5.4 \times 10^{-7}$) |
| Chongqing | Chongqing | 4151 (3434, 4871) | 0.0494 (0.0444, 0.0529) | 1.3322 (1.1976, 1.4272) | $2.0 \times 10^{-6}$ ($1.4 \times 10^{-6}$, $2.6 \times 10^{-6}$) |
| Yunnan | Baoshan | 163 (101, 314) | 0.0313 (0.0285, 0.0348) | 1.7942 (1.6355, 1.988) | $1.5 \times 10^{-7}$ ($2.3 \times 10^{-9}$, $1.2 \times 10^{-6}$) |
| | Pu'er | 901 (661, 1203) | 0.0401 (0.0377, 0.0453) | 2.3125 (2.1769, 2.6038) | $3.3 \times 10^{-6}$ ($1.6 \times 10^{-6}$, $6.7 \times 10^{-6}$) |
| | Lincang | 1002 (763, 1399) | 0.0415 (0.0381, 0.046) | 2.4552 (2.2579, 2.7164) | $7.0 \times 10^{-6}$ ($3.8 \times 10^{-6}$, $1.2 \times 10^{-5}$) |
| | Honghe Hani and Yi Autonomous Prefecture | 611 (472, 850) | 0.0421 (0.0387, 0.0482) | 2.4061 (2.2163, 2.7503) | $1.7 \times 10^{-6}$ ($9.9 \times 10^{-7}$, $3.2 \times 10^{-6}$) |
| | Wenshan Zhuang and Miao Autonomous Prefecture | 34 (19, 63) | 0.0348 (0.0316, 0.0395) | 2.0064 (1.8242, 2.274) | $1.8 \times 10^{-8}$ ($2.7 \times 10^{-11}$, $2.2 \times 10^{-7}$) |
| | Xishuangbanna Dai Autonomous Prefecture /Sipsongpanna | 13753 (11240, 17205) | 0.0502 (0.0469, 0.0558) | 2.9377 (2.7562, 3.2702) | $1.5 \times 10^{-4}$ ($1.0 \times 10^{-4}$, $2.2 \times 10^{-4}$) |
| | Dehong Dai and Jingpo Autonomous Prefecture | 4860 (3882, 6264) | 0.0479 (0.0448, 0.0533) | 2.8391 (2.6613, 3.1603) | $5.8 \times 10^{-5}$ ($3.9 \times 10^{-5}$, $8.9 \times 10^{-5}$) |
| | Nujiang Lisu Autonomous Prefecture | 2 (0, 10) | 0.0299 (0.0265, 0.0335) | 1.7044 (1.5061, 1.9158) | $4.1 \times 10^{-12}$ ($3.6 \times 10^{-12}$, $3.0 \times 10^{-7}$) |

Estimated total infections include reported infections and unreported infections for a full year. The average of daily results include transmission rate, effective reproduction number and force of infection in September and October. The set median and 95% CIs are displayed outside and inside the brackets, respectively.

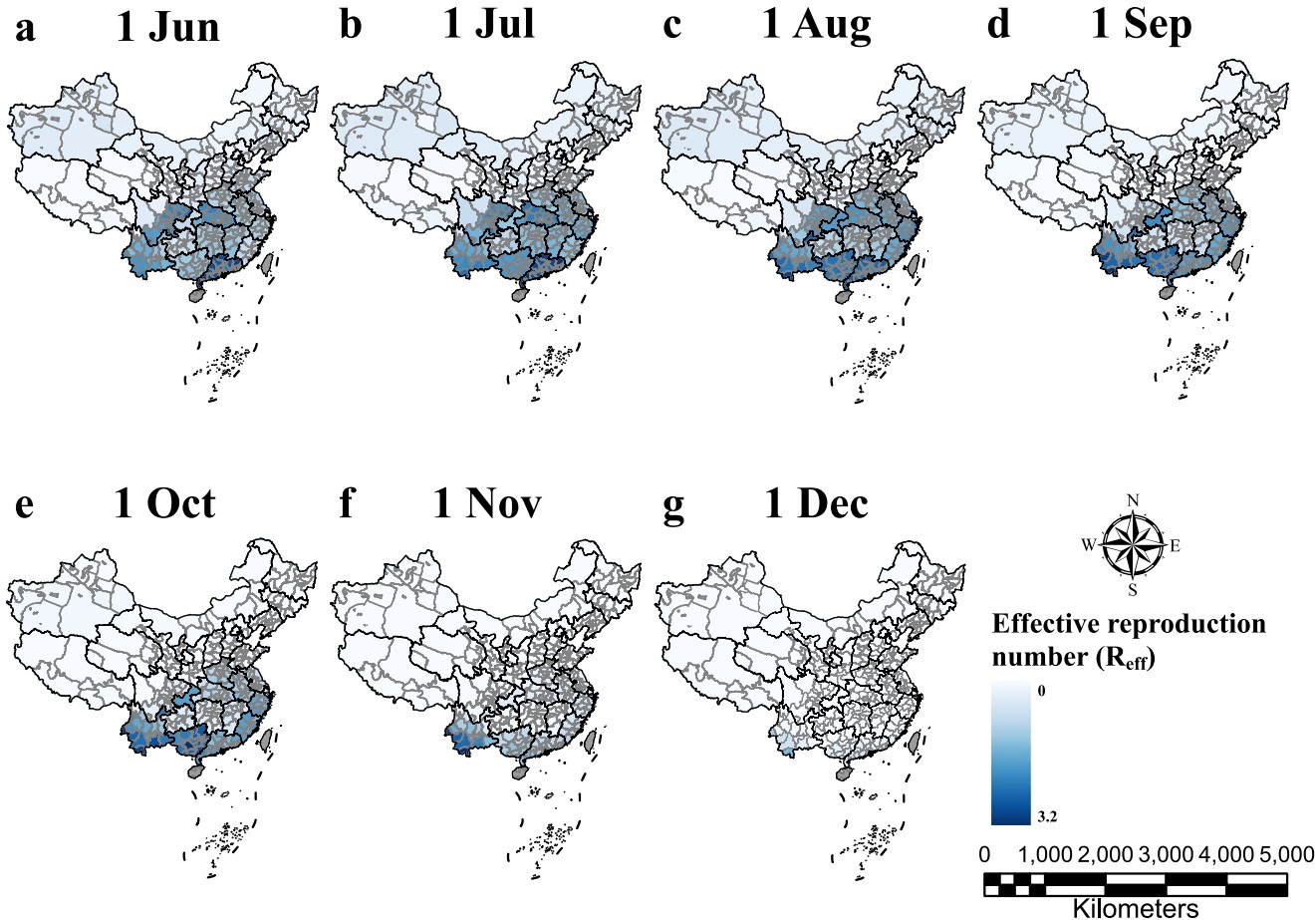

**Fig. 5 | Estimates of effective reproduction numbers in cities.** (**a**–**g**) Estimates of effective reproduction numbers for dengue on the first day of each month in each city. Blue shades represent the magnitude of the effective reproduction number, while gray shades represent the absence of inference.

several major urban clusters[31]. In each of the major urban agglomerations, we estimated that the number of infections peaks in September or October, which was broadly in line with our previous studies[12,13]. In addition, similar trends in the $R_{eff}$ estimated by the model revealed that dengue fever gradually tends to subside around October. The identification of peak timing and the $R_{eff}$ played a vital role in targeting measures to prevent and control dengue outbreaks. As a key indicator, changes of the $R_{eff}$ in cities across the country were also demonstrated. Our research revealed that the cities with high $R_{eff}$ were mainly concentrated in southern China, especially in some border and coastal cities. Despite the incomplete consistency of the mathematical models used, the estimates of $R_{eff}$ in different regions in our study are consistent with the results of some previous studies in partial regions of China[32–35]. Dengue burden surveys helped to elucidate the temporal distribution of infected cases, which is likely attributed to factors such as particular temperature and high humidity[27,36,37], and these in turn had informed the spatial distribution of dengue. As data from more years and regions become available in the future, we hope to further validate and refine the temporal, spatial and population distributions of dengue fever.

Another potentially important trigger for the incidence of dengue fever was the increased mobility of the population[13,38,39]. As confirmed by our study, dengue fever spread with a lag in China. Dengue fever in China occurs in coastal and border cities from August to October, and gradually spreads inland. The results of population movement partially underpinned this trend. Population movements could lead to infections mainly in the southern pivotal cities with high population movements and radiate to around cities. These findings might be

driven by a number of factors that regulate population mobility, including national policies and economic developments[40,41]. Further research is needed on this preliminary finding to alleviate the burden of dengue fever.

On the basis of the knowledge of the transmission mechanism of DENV depicted by a set of ODEs[12], the mathematical model consisting of both mosquito and human compartments was developed to simulate dengue transmission in this study. We further incorporated the transmission dynamics model with population movement, and then constructed the city-resolved metapopulation network system to simulate the transmission of DENV between cities of China. An iterative Bayesian inference algorithm-the EAKF was applied to assimilate observations of daily dengue cases in each city, as well as population movement between cities. Our previous study demonstrated that the unobserved state variables (for example, new infections per day within a city) and system parameters (for example, transmission rate in each city) of the model can be iteratively calibrated and estimated by a fitting of the model to case observations using Bayesian inference[13,42]. Our results showed that the method can recover key time-varying transmission parameters of DENV in multiple simulation scenarios. The model fitting to the observed dengue case data accurately captured the successive epidemic waves occurring at national and city scales.

Some limitations in our study should be noted. First, although we used real measurements from 89 mosquito vector surveillance sites in our analysis, the current surveillance of mosquito vector density in some remote areas is not comprehensive, resulting in possible biases in the vector density estimates obtained. Second, this study examined

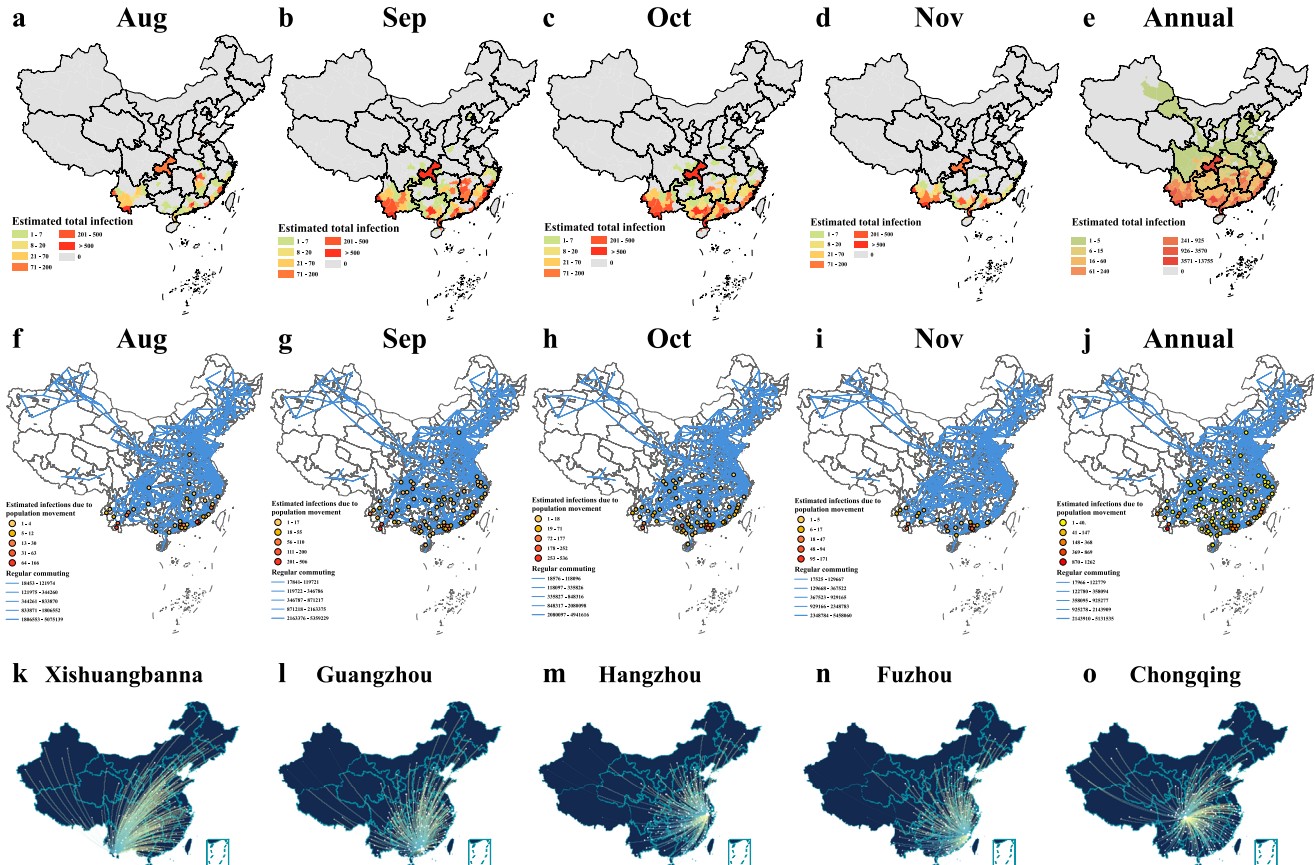

**Fig. 6 | Estimates of dengue infections in cities and the relationship between population mobility and dengue infections.** (**a–e**) Estimates of the number of dengue infections in each city for each month and for the whole year. Redder colors in the cities indicate more severe infections, and the legends record the infections' range of specific numbers. (**f–j**) Estimated infections due to population movement in each city and the network of population movement intensity between cities. Estimated infections attributed to population movement in each city, along with a network illustrating the intensity of population movement among cities. Dots represent infections in cities. The thickness of the blue line represents the intensity of population movement between cities. (**k–o**) Population outflows in five cities. The lines and dots indicate the location of the population outflow from the city. Larger pink dots indicate higher population outflows. Note that the South China Sea part in the figure k-o is consistent with the previous figure, which is not shown due to the limited space in the figure.

the spread of dengue fever based on large-scale population movements between cities. However, the study was unable to determine the subtle differences between the actual distance individuals traveled, the amount of time spent in different locations, and the frequency of commuting by populations in specific types of habitats (e.g., residential, recreational, and commercial facilities)[13]. Third, population movement is one of the major triggers for cross-regional transmission of dengue fever. Although we considered cross-regional transmission within the country, international population movement could not be well estimated and applied in the model. It is hoped that the effects caused by population mobility can be elaborated in more detail in further studies. Finally, although our model takes into account the heterogeneity of disease transmission among populations in different regions (e.g., dengue transmission capacity, and so on), the study did not take into account the different effects such as classifications of primary and post-primary dengue cases and cases with different dengue severity on disease transmission since the information is not yet available. Although the data of dengue fever cases used in our study were obtained from the NNDSS, currently the largest national surveillance platform of infectious diseases in China, the influence of misclassification of dengue cases on the estimated results still cannot be totally eliminated.

In conclusion, this study gave a comprehensive quantification of the time-evolving epidemiological features and burden associated with successive epidemics waves of dengue fever in China over recent years. Our study offers robust, population based, national data to understand epidemiological characteristics and transmission dynamics of dengue fever. These efforts will facilitate the transformation of dengue transmission inference into decision-making. Future mitigation policies should take into account the heterogeneity of the disease burden found.

## Methods

### Ethics statement

This study complies with all relevant ethical regulations. The approval was obtained from the Ethics Committee of National Institute Communicable Disease Control and Prevention Chinese Center for Disease Control and Prevention (Ethical Approval NO. ICDC-2016008).

### Study domain

In this study, the unit of data analysis was defined as a prefecture-level administrative region or municipality (hereunder referred to uniformly as cities). A total of 333 prefecture-level administrative regions (including 293 prefecture-level cities, 30 autonomous prefectures, 7 prefectures and 3 leagues) were included in our analysis. We also included 4 municipalities (including Beijing, Shanghai, Tianjin, and Chongqing) in our analysis because dengue fever was active in these places. Therefore, all 337 cities in mainland China were included in this study. Additionally, we divided these 337 cities with severe dengue outbreaks into major urban agglomerations (Supplementary pp 20).

## Dengue fever incidence data

Surveillance data on daily incidence of dengue fever during the period from 1st January, 2013 to 31st December, 2020 were obtained from the National Notifiable Diseases Surveillance System (NNDSS) of the Chinese Center for Disease Control and Prevention (China CDC). All dengue fever cases were diagnosed according to the diagnostic criteria for Dengue Fever (WS 216-2008 and WS 216-2018)[43,44]. Case definition for a probable, clinically diagnosed and laboratory-confirmed case of dengue fever disease can be found in the appendix document (Supplementary pp 6). The comprehensive diagnosis was made according to the epidemiological evidence, clinical manifestations and laboratory examination results of the patients. The information of dengue fever cases includes basic demographic information (gender, age, date of birth, national standard code of individual present residential address, and population classification), disease diagnosis (probable cases, clinically diagnosed cases, and laboratory-confirmed cases), date of disease onset, date of diagnosis, and date of death (if applicable). The daily dengue fever cases were aggregated to each city based on individual present residential address.

## Mosquito abundance data

Two types of mosquito abundance data were simultaneously collected specifically for this study and utilized to evaluate mosquito larvae and adult mosquito density in each region. Mosquito adult density (Aedes aegypti and Aedes albopictus) was sampled by mosquito light traps. In high-risk areas with frequent dengue outbreaks in recent years, mosquito vector activity is monitored twice a month. In areas where local cases have occurred in recent years or where the risk of dengue outbreaks is relatively high, mosquito vector activity is monitored once a month. The number of adult mosquitoes was evaluated in several circumstances, such as agricultural areas, livestock sheds, residential areas (> 50 households sampled per month), and parks. Breteau index (BI) was calculated to evaluate the mosquito larvae density[45]. $BI = N^*/N \times 100$ evaluates the breeding of mosquito larvae in indoor and outdoor small water containers. $N^*$ is the number of Aedes positive water storage containers, and $N$ is the total number of households surveyed (> 50 households sampled per month). All operation was performed according to the standard of National Vector Detection Program[46]. The locations of 89 survey sites for mosquito vector data are shown in Supplementary Fig. 8.

## Meteorological surveillance data

Daily meteorological data including ambient temperature, daily maximum temperature and daily minimum temperature from 1st January, 2013 to 31st December, 2019 were obtained from the China Meteorological Data Sharing System (http://data.cma.cn/). This dataset comprises information collected from 2441 meteorological stations in mainland China, and organized based on their respective administrative regions (Supplementary Fig. 8). In cases where multiple meteorological stations are available within a single administrative region, the regional average, maximum, and minimum temperature were calculated as the average, maximum, and minimum values of multiple stations, respectively.

## Demographic and population mobility data

Demographic and population mobility data was collected from the results of China's seventh census in 2020, which can be publicly obtained from the National Bureau of Statistics of China (http://www.stats.gov.cn/). In order to match this outflow and inflow of population to our prefecture-level administrative and to obtain a more accurate and detailed understanding of the daily population commute, we obtain daily city-level population migration data from the Gaode Migration Big Data platform (https://trp.autonavi.com/migrate/page.do). The platform of Gaode Migration Big Data is based on location-based service (LBS), which collects the

geographical location changes of users' mobile phone positioning to depict the migration intensity between cities. To ensure privacy and confidentiality, the Gaode platform only provides the weight parameter of the daily flow scale between cities, named migration index. Hence, we calculate the daily inflow and outflow population between city by $n_{A \leftarrow B}(t) = \eta_{A \leftarrow B}(t) / \sum_{k \neq A} \eta_{A \leftarrow k}(t) \times \rho_A$. Here, $n_{A \leftarrow B}(t)$ is the migration index from city $B$ to city $A$ on day $t$. The $\rho_A$ is a static inflow population of city $A$ from the census data. Noted that since the platform was opened around June 2018, to reduce the impact of the COVID-19 pandemic on population mobility patterns, we selected the migration index for the whole year from 1st June, 2018 to 31st May, 2019 to match each calendar year. We assume that the population mobility patterns of each calendar year during the study period were similar.

## Descriptive analysis of local and imported dengue cases

We included all cases of dengue fever from 1st January, 2013 to 31st December, 2020 in our analysis. We categorized all cases into local and imported cases and assessed the temporal and spatial geographic distribution in 31 provincial administrations. We categorized cities into 4 × 4 categories according to the number of local and imported cases in each city by using the method of Jenks natural discontinuity classification[47], and a map was generated. To quantify the seasonal pattern of local dengue cases in each region, we plotted heat maps of dengue cases throughout the year and on a weekly basis for local cases of dengue. We also stratified local cases by different diagnostic categories. We stratified imported cases by original country and imported Chinese provincial administrations to map the source-sink relationship of dengue imported cases.

## Latitudinal gradient characteristics of dengue cases distribution

We also fitted models of seasonal multiple linear regression to weekly time series of local and imported dengue cases (Supplementary pp 11), including time trends and harmonic terms to represent annual and semiannual periodicity[48-50]. We extracted the amplitudes and peak times of the annual and semiannual cycles based on the model coefficients[48]. The amplitude measures the difference between the maximum and minimum values of the epidemic curve for the disease[50]. The $F$-test method was applied to explore the statistical significance of seasonal multiple linear regression models and determine whether the overall linear relationship holds.

## Metapopulation network modeling for local dengue cases

We developed a metapopulation network model focusing on simulating the spread of dengue fever at prefecture city level in mainland China in 2019. This model extends the traditional isolated compartment model by treating each city as a subpopulation and interconnecting then through population movement, allowing for superior performance even when the internal space does not satisfy the assumption of population homogeneity. It has shown superiority in tracking the large-scale spatial spread of infectious disease[12,13,42].

Dengue fever is an infectious disease transmitted by mosquitoes, primarily Aedes aegypti and Aedes albopictus. DENV is represented by four distinct serotypes (DENV-1 to DENV-4) that co-circulate in tropical and sub-tropical regions and can repeatedly infect humans. Transmission processes are divided into two steps: infected individuals infect susceptible mosquitoes, then infected mosquitoes infect susceptible individuals. This assumption is based on evidence that infection with any serotype of DENV leads to a short period of heterotypic immunity (that is, immunity against another serotype)[51] and the reality that four DENV serotypes are commonly transmitted across different regions in mainland China[52]. Therefore, a mathematical model consisting of both mosquito and human compartments (SI-SIR model) is considered in our study to simulate dengue transmission. Susceptible

($S_H$), infected ($I_H$) and removed ($R_H$) humans are included in the human compartment.

The mosquito compartments consist of both susceptible ($S_M$) and infected ($I_M$) mosquitoes. This model has been validated in previous studies[12,13]. The SI-SIR model comprises the following system of ordinary differential equations (ODEs):

$$\frac{dS_M}{dt} = -\frac{\lambda\tau(t)\beta_M S_M I_H}{N_H} - \alpha S_M + \mu_b(t)(S_M + (1-U)I_M) - \mu_d S_M$$

$$\frac{dI_M}{dt} = \frac{\lambda\tau(t)\beta_M S_M I_H}{N_H} + \alpha S_M + \mu_b(t)U I_M - \mu_d I_M$$

$$\frac{dS_H}{dt} = -\frac{\lambda\tau(t)\beta_H S_H I_M}{N_H}$$

$$\frac{dI_H}{dt} = \frac{\lambda\tau(t)\beta_H S_H I_M}{N_H} - \frac{I_H}{D}$$

Here, $N_H$ represents the total population number; $\lambda$ is the adjusted transmission rate between humans and mosquitoes; $\tau(t)$ is the biting rate; $\beta_M$ is the probability for the infection transmission from human to mosquitoes; $\beta_H$ is the probability for the infection transmission from mosquitoes to human; $\alpha$ is the random rate of DENV seeding into mosquitoes; $\mu_b(t)$ is the birth rate of mosquitoes at time $t$; $\mu_d$ is the mortality rate of mosquitoes; $U$ is the vertical dissemination rate of infected mosquitoes and is constant over an outbreak; $D$ is the average duration of contagiousness for human.

To use the compartment model across the national cities, it is necessary to take into account population movements between cities, for which we extend the isolated model to a metapopulation model. In this extended model, we included two types of population movements, daily regular commuting and diffusive random movements across the 337 cities in China[42]. The daily regular commuting is relatively stable in terms of demographic composition, and travel regularly between places of residence and work. In contrast, the diffusive random movements circulate among all subpopulations, following a Markov process specified by a transition matrix. Previous studies have shown that these two patterns of population movement have qualitatively different effects on the spatial spread of infectious diseases[53,54]. We therefore take both patterns of population movement into account.

The population exchange pattern by regular commuting is defined as follows. During the daytime, the population in location $i$, $N_i^{Hd}(t) = N_{ii}^H + \sum_{k \neq i} I_{ki}^{Hr}(t) + \sum_{k \neq i}(N_{ik}^H - I_{ik}^{Hr}(t))$, consists of three parts: individuals who live and work in location $i$; reported infected individuals who would otherwise move to location $k(k \neq i)$; unreported infected individuals who work in location $i$. For the subpopulation $N_{ij}$, the new infected mosquito come from both the reported and unreported infections in location $i$. For each susceptible mosquito $S_i^M$ in population $N_i$, the probability of contacting reported infected individuals is $\sum_k I_{ki}^{Hr}(t)/N_i^{Hd}(t)$, where $\sum_k I_{ki}^{Hr}(t)$ is the reported infected individuals who would otherwise move to location $k(k \neq i)$, and the probability of contacting unreported infected individuals is $\sum_k I_{ik}^{Hu}(t)/N_i^{Hd}(t)$, where $\sum_k I_{ik}^{Hu}(t)$ is the unreported infected individuals who work in location $i$. Those contacts lead to new infected mosquito $\frac{\lambda_i(t)\tau_i(t)\beta_i^M(t)S_i^M(t)\sum_k(I_{ki}^{Hr}(t)+I_{ik}^{Hu}(t))}{N_i^{Hd}(t)}$ during a period of $dt_1$ day. For each susceptible individuals $S_{ij}^H(t)$ in population $N_i$, the probability of contacting infected mosquito is $I_i^M(t)/N_i^{Hd}(t)$. Those contacts lead to new infected individuals $\frac{\lambda_i(t)\tau_i(t)\beta_i^H(t)S_{ij}^H(t)I_i^M(t)}{N_i^{Hd}(t)}$ during a period of $dt_1$ day.

The population exchange pattern by random movement is defined as follows. Apart from regular commuting, during the daytime, $\theta dt_1 \bar{N}_{ik}^H$ individuals (except for reported infected individuals) drawn from the location $k(k \neq i)$ move to location $i$, are randomly redistributed into the subpopulation here. This population exchange pattern exists between any two locations. For instance, we compute the number of susceptible individuals entering into subpopulation $S_{ij}^H(t)$. In other location $k(k \neq i)$, the probability that a random traveler is susceptible is $\sum_l S_{kl}^H(t)/(N_i^{Hd}(t) - \sum_l I_{lk}^{Hr}(t))$, where the numerator $\sum_l S_{kl}^H(t)$ is the number of susceptible individuals from all other locations $l$ to the location $k$, and the denominator $N_k^{Hd}(t) - \sum_l I_{lk}^{Hr}(t)$ is the total mobile population (total population minus reported infected individuals) in location $k$. Therefore, the total number of susceptible individuals entering location $i$ is $\theta dt_1 \sum_{k \neq i} \bar{N}_{ik}^H \sum_l S_{kl}^H(t)/(N_k^{Hd}(t) - \sum_l I_{lk}^{Hr}(t))$. These individuals are redistributed into subpopulations present in location $i$, where the proportion of population in subpopulation $N_{ij}$ is $(N_{ij}^H - I_{ij}^{Hr}(t))/N_i^{Hd}(t)$. The number of susceptible individuals entering the subpopulation $S_{ij}^H(t)$ is calculated by $\theta dt_1 \frac{N_{ij}^H - I_{ij}^{Hr}(t)}{N_i^{Hd}(t)} \sum_{k \neq i} \frac{\bar{N}_{ik}^H \sum_l S_{kl}^H(t)}{N_k^{Hd}(t) - \sum_l I_{lk}^{Hr}(t)}$.

In addition, we calculate the number of susceptible individuals leaving the subpopulation $S_{ij}^H(t)$. The $\theta dt_1 \sum_{k \neq i} \bar{N}_{ki}^H$ individuals are redistributed leaving location $i$. The proportion of susceptible people from $N_{ij}$ is $S_{ij}(t)/(N_i^{Hd}(t) - \sum_l I_{li}^{Hr}(t))$. Finally, the number of susceptible persons leaving the subpopulation $S_{ij}(t)$ is calculated by $\theta dt_1 \frac{S_{ij}^H(t)}{N_i^{Hd}(t) - \sum_l I_{li}^{Hr}(t)} \sum_{k \neq i} \bar{N}_{ki}^H$.

Random movement for other human compartments can be computed similarly. Note there is no random movement in reported infected individuals $I_{ii}^{Hr}(t)$ and mosquito compartments because we assume infections and mosquitoes were not able to move long distance. Similarly, we can also describe the human and mosquito transfer process at nighttime.

These regular commuting and random movement only occur between human populations, by comparison, mosquitoes generally don't move long distances. Therefore, this study assumed that the mosquitoes are fixed in a single city from birth to death. We also distinguish between reported infected and unreported infected individuals, and defined different movement patterns for them. To better fit real case observations as well as unobservable estimates of state variables and system parameters, we coupled a city-resolved metapopulation model with an iterative Bayesian inference algorithm, named the EAKF[55]. The model is able to depicts both reported and unreported infections and recover key epidemiologic time-varying parameters[12,13,56–59]. The specific details of the extended metapopulation network model are provided in the Supplementary Information pp12-15.

## Transmission dynamics and parameters of dengue fever

The transmission model estimates the total infections (both reported and unreported) in China and the four major urban agglomerations. The estimated infection due to population movements is calculated as follows: $I_{move} = I^{Hr} + I^{Hu} - I_{ii}^{Hr} - I_{ii}^{Hu}$, which is simply the total infectious in each city minus the infectious caused by local commuters. The transmission rate $\beta_i(t)$ from mosquitoes to human in location $i$ can be estimated by $\beta_{M \to H}(t) = \lambda_i(t)\tau_i(t)\beta_i^H(t)$. Similarly, the transmission rate from humans to mosquitoes $\beta_{H \to M}(t) = \lambda_i(t)\tau_i(t)\beta_i^M(t)$.

For a constant transmission rate, the basic reproduction number ($R_0$) derived from the next generation matrix[60] is estimated by the formula $R_0 = \sqrt{\frac{\beta_{M \to H}\beta_{H \to M}N_M D}{N_H(\mu_d - \mu_b U)}}$. As time evolves, the basic reproduction number $R_0$ cannot track the progress of the transmission because the susceptible individuals and mosquitoes ($S_H$ or $S_M$) are gradually consumed, become infected and finally removed from the disease transmission cycle. Therefore, the effective reproduction number ($R_{eff}$) was

defined to track the evolution of the indicator[61] in the following equation

$$R_{eff} = R_0 \sqrt{\frac{S_i^H(t)}{N_i^H} \frac{S_i^M(t)}{N_i^M}} = \sqrt{\frac{\lambda_i^2(t)\tau_i^2(t)\beta_i^M(t)\beta_i^H(t)S_i^M(t)S_i^H(t)D}{(\mu_d - \mu_b(t)U)(N_i^H(t))^2}},$$

where $S_i^H(t) = S_{ii}^H(t) + \sum_{k \neq i} S_{ik}^H(t)$ is the number of susceptible individuals in location $i$; $N_i^H(t) = N_{ii}^H + \sum_{k \neq i} I_{ki}^{Hr}(t) + \sum_{k \neq i}(N_{ik}^H - I_{ik}^{Hr}(t))$ is the number of populations in location $i$. For this calculation, $S_i^H(t) = N_i^H(t)$ is assumed because the infected individuals are less than 3‰ populations in each location, and then the indicator $R_{eff}$ can be calculated by the following equation

$$R_{eff} = R_0 \sqrt{\frac{S_i^H(t)}{N_i^H} \frac{S_i^M(t)}{N_i^M}} = \sqrt{\frac{\lambda_i^2(t)\tau_i^2(t)\beta_i^M(t)\beta_i^H(t)S_i^M(t)D}{(\mu_d - \mu_b(t)U)S_i^H(t)}},$$

where $R_{eff}$ represents the average number of generations of infected people caused by each imported or local case on a given day in each city.

Furthermore, we defined the force of infection ($F_i(t)$) based on this transmission model $F_i(t) = \frac{\lambda_i(t)\tau_i(t)\beta_i^H(t)I_i^M(t)}{N_i^H(t)} = \frac{\lambda_i(t)\tau_i(t)\beta_i^H(t)I_i^M(t)}{S_i^H(t)}$. This indicator $F_i(t)$ represents the probability of a susceptible person becoming infected with dengue, depending on the likelihood that the susceptible person acquires the DENV from an infected mosquito. Note that the above $R_{eff}$ and $F_i(t)$ ignore the influence of random movement.

## Reporting summary

Further information on research design is available in the Nature Portfolio Reporting Summary linked to this article.

## Data availability

Dengue surveillance data are available at the National Notifiable Disease Surveillance System of the Chinese Center for Disease Control and Prevention (China CDC). To access these data and/or to seek permission for its use, please contact the Data-center of China Public Health Science (https://www.phsciencedata.cn/Share/index.jsp) or email data@chinacdc.cn. Mosquito vector density data is subject to restrictions for the protection of privacy. Requests for data access should be addressed to the corresponding authors. All other data utilized in this study were accessed from open sources. Meteorological data was obtained from the China Meteorological Data Sharing System (http://data.cma.cn/). Demographic and mobility data was collected from the China's seventh census in 2020, which can be publicly obtained from the National Bureau of Statistics of China (http://www.stats.gov.cn/). The population migration data at daily city-level was obtained from the platform of Gaode Migration Big Data (https://trp.autonavi.com/migrate/page.do).

## Code availability

Custom codes supporting the statistical analysis are publicly available at GitHub (https://github.com/tinkzzz/DENGUE_CHINA.git) and Zenodo[62].

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

## Acknowledgements

The study was funded by the State Key Laboratory of Infectious Disease Prevention and Control (Chinese Center for Disease Control and Prevention) (No. 2021SKLID502 received by X.B.L.). The funder had no role in study design, data collection and analysis, decision to publish, or preparation of the manuscript. We also appreciate the efforts of the staff at the hospitals, local health departments, and municipal-, provincial-, and national- level Centers for Disease Control and Prevention in the data collection process.

## Author contributions

H.B.N., Q.Y.L., X.B.L., and P.G. conceived and designed the study. H.B.N., X.Y.C., J.R.R., T.T.D., J.Y.Z., J.M.L., L.W., Y.C.Y., T.X., L.N.X., L.X.W., X.B.L., and P.G. performed the research. H.B.N., X.Y.C., J.R.R., L.W., S.P., X.B.L., and P.G. carried out the data analyzes. H.B.N., X.Y.C., X.B.L., and P.G. wrote the paper. P.G., X.B.L., Q.Y.L., and S.P. edited the paper.

## Competing interests

The authors declare no competing interests.
