## [Peer Review File · Nature Communications]

Reviewers' Comments:

Reviewer #1:

Remarks to the Author:

In this study the authors fitted weekly dengue time series and estimated the amplitude and peak timing of the annual and semiannual cycles using nationwide data of dengue cases, surveillance data for mosquito density and population mobility. Then a data-driven model-inference approach was used to simulate the epidemic at city-scale and estimate time-evolving epidemiological parameters. The authors found that the geographical distribution of dengue cases was expanding, and the main imported areas as well as external sources of imported cases changed; the annual amplitude of dengue epidemic varied with latitude; the effective reproduction number across cities is commonly greater than 1. I think this is an important and interesting topic. Generally, the paper is well written and strategically constructed to enlighten readers. It will be acceptable after addressing the following minor issues.

1. Reference 3 is not the best one to describe the fact. I would suggest the authors find another literature.
2. Line 114-116: The authors claimed that “However, the overall burden and spatiotemporal spread characteristics of dengue fever in China have not been comprehensively quantified.” As I know, there are some studies focusing on the national burden of dengue in China. I would suggest the authors to describe some similar studies and point out the research gaps.
3. Line 138: This study include 337 cities in mainland China, but readers may not know the total number of cities in mainland China. Letting readers know the coverage of study areas in this paper would be better.
4. Line 157-163: for the mosquito vector density data, the authors described the details of monitoring vector density, but missed one important information, i.e., the frequency of sampling.
5. Line 171-174: Daily meteorological data were obtained from 2013 to 2019, but the study period was 2013 to 2020. Why?
6. Line 446-447: “We found that the variation of R_{eff} for dengue fever in cities across the nation”. This sentence is confusing.
7. Line 473-475: the authors claimed that “Our study of more than 90,000 cases of dengue fever reported to the NNDSS during 2013-2020 in China gives the most comprehensive quantification of national burden and time-evolving intrinsic dynamic characteristics of the disease up to now.” Use of strong words such as “the most” should be careful.
8. What are the limitations of this study? There should be a limitation section.

Reviewer #2:

Remarks to the Author:

In this study, Haobo and colleagues performed a large-scale statistical and modeling study on the dengue transmission dynamics in China. They combined nationwide data on around 100K dengue cases, 89 surveillance sites for mosquito density, and population mobility between 337 cities over the period 2013-20. Their statistical and modeling analyses identified several interesting findings, including the geographic expansion of dengue distribution and seasonality and key drivers of dengue spread in China. Their research outputs could be useful to inform local dengue control policies.

Several major comments:

1. In supplementary lines 150 to 277, the authors had two duplicate sections of dengue diagnostic methods, with inconsistent details. Could you provide a summary table of probable, clinical diagnosed, and lab-confirmed diagnostic standards?

2. For laboratory diagnosis, many important information has been missing. For example, the protocol of IgM, IgG dengue serology, RT-PCR, virus isolation, and NS1 antigen test. How about the case misclassification rate for laboratory, clinical and epidemiological diagnosis? Proportion of primary and post-primary dengue cases in local / imported cases? Proportion of cases with different severity (i.e., dengue fever, dengue haemorrhagic fever, dengue shock syndrome).

The authors could discuss the influence of these case definition factors on the estimated results.

3. In section “Metapopulation network modelling for local dengue cases” in Main Text, the SI-SIR model might be oversimplified for modeling dengue transmission between mosquitoes and human populations. The authors could use the SEI-SEIR model framework [1] that addresses the mosquito life-history dynamics and dependency of virus development rate to temperature.

4. It would be helpful if the authors could find some evidence or literature to support the assumption that the human populations remain fully susceptible to dengue infection in China.

5. In Table 1, the average of daily effective reproduction number is estimated to be larger than 1 in each considered city and province. This may suggest that dengue has been endemic in these cities in 2019?

However, the average force of infection is estimated to be very low, which might be

inconsistent with the estimates of R_{eff} . What's the average generations of infected people caused by each imported / local case?

6. In supplementary Fig. 30, the authors analyzed each year from 2013 to 2018. However, the main text seems mainly taking about the results for 2019. The authors may wish to discuss the interannual variations and corresponding drivers.

Refs.

[1] Climate predicts geographic and temporal variation in mosquito-borne disease dynamics on two continents. *Nat. Commun.* 12, 1233 (2021).

Reviewer #3:

Remarks to the Author:

This is an overall welcome manuscript describing the burden of dengue in China, a country that has to date not been as affected by dengue compared to South East Asia and Latin America. The authors employed a data-driven model-inference approach to simulate the dengue incidence at city-scale and estimate time-evolving epidemiological parameters. The higher incidence in Southern cities compared to Northern cities is not surprising, and not new. What is interesting is the older age group that is affected compared to what we see in truly dengue endemic countries. Several language issues throughout the manuscript which can be fixed with some editorial help.

Abstract :

Avoid terms such as « super outbreaks ». No outbreak in China has been as large as the one we are currently seeing in Brazil, with 4 million cases within a matter of January till April 2024, or outbreaks such as seen in SE Asia over the past decades. “Super” is therefore misleading, especially as the next sentence says 95,000 cases. That is truly a small number compared to the 100 million cases worldwide and compared to the large population size of China.

What do you mean by your statement “it suggested that dengue did not seem to become an endemic disease at present”. Do you mean it is an epidemic disease? It is already endemic!

Introduction: How can the authors call 40,000 cases a “super outbreak”? The authors also fail to state that urbanization and increasing population densities are the main drivers for dengue, not climate change. Obviously, climate and weather parameters are

important for the vector, and indeed the vector can expand to naïve areas with climate change. However, in areas where the ecological conditions are already favorable to the vector, the increasing incidence of dengue is due to increasing population densities.

The authors would need to check whether increasing population size in the Southern Cities are correlated with the increasing incidence.

References:

Several references are old (eg Ref 1, 2, 6, 7 etc) and should be updated. There are also some other references related to China that the authors could add.

Here some suggested references:

Global burden for dengue and the evolving pattern in the past 30 years.

Yang X, Quam MBM, Zhang T, Sang S. *J Travel Med.* 2021 Dec 29;28(8):taab146. doi: 10.1093/jtm/taab146. PMID: 34510205

Discrepancies in dengue burden estimates: A comparative analysis of reported cases and Global Burden of Disease Study, 2010-2019.

Lee SY, Shih HI, Lo WC, Lu TH, Chien YW. *J Travel Med.* 2024 May 2:taae069. doi: 10.1093/jtm/taae069. Online ahead of print. PMID: 38696416

Reviewing estimates of the basic reproduction number for dengue, Zika and chikungunya across global climate zones.

Liu Y, Lillepold K, Semenza JC, Tozan Y, Quam MBM, Rocklöv J. *Environ Res.* 2020 Mar;182:109114. doi: 10.1016/j.envres.2020.109114. Epub 2020 Jan 3. PMID: 31927301

Global burden and trends of neglected tropical diseases from 1990 to 2019.

Lin Y, Fang K, Zheng Y, Wang HL, Wu J. *J Travel Med.* 2022 May 31;29(3):taac031. doi: 10.1093/jtm/taac031. PMID: 35238925

The Lancet Commission on dengue and other Aedes-transmitted viral diseases.

Wilder-Smith A, Lindsay SW, Scott TW, Ooi EE, Gubler DJ, Das P. *Lancet.* 2020 Jun 20;395(10241):1890-1891. doi: 10.1016/S0140-6736(20)31375-1.

Increasing Dengue Incidence in Singapore over the Past 40 Years: Population Growth, Climate and Mobility.

Struchiner CJ, Rocklöv J, Wilder-Smith A, Massad E. *PLoS One.* 2015 Aug 31;10(8):e0136286. doi: 10.1371/journal.pone.0136286. eCollection 2015. PMID: 26322517

To Reviewer 1:

Dear professor, thank you very much for your encouragement “I think this is an important and interesting topic. Generally, the paper is well written and strategically constructed to enlighten readers. It will be acceptable after addressing the following minor issues”. You have given us the excellent feedbacks that helped improving the manuscript and sparked ideas for our future studies. We really thank you for the insightful and constructive comments, although you feel that some issues should be clarified to improve the manuscript. We have carefully revised the manuscript based on your comments. We hope these changes will make it easier for potential readers to understand. Our responses to the comments have been arranged one by one in the following section. Please check them.

1) Comment: "Reference 3 is not the best one to describe the fact. I would suggest the authors find another literature."

Response: Dear Professor, we really appreciate your valuable suggestion. We really agree with you that another literature should be provided to help describe the fact better. Therefore, we have updated the reference to more appropriately support the statement (please refer to: *Kraemer MUG, Reiner RC Jr, Brady OJ, et al. Past and future spread of the arbovirus vectors Aedes aegypti and Aedes albopictus. Nat Microbiol. 2019;4(5):854-863*). Please check the corresponding revision in **the References part of the revised manuscript**. Thank you.

2) Comment: "Line 114-116: The authors claimed that "However, the overall burden and spatiotemporal spread characteristics of dengue fever in China have not been comprehensively quantified." As I know, there are some studies focusing on the national burden of dengue in China. I would suggest the authors to describe some similar studies and point out the research gaps."

Response: Dear Professor, thank you very much for this insightful comment and your

rigorous attitude. We apologize for this unreasonable statement in the version of the manuscript submitted in the first round. We really agree with you that there are some studies focusing on the national burden of dengue in China, however we find that these studies mainly focused on identifying the connectivity between local and imported cases as well as the fundamental factors (climate drivers and so on) for dengue epidemics. The knowledge on the time-evolving transmission dynamic features and burden associated with nationwide spread of dengue fever in China is still limited. Therefore, this present study aims to address this research gap. According to your important suggestions, we have described some similar studies and pointed out the research gaps. Please check in **the 2nd paragraph of the Introduction part**. Thank you very much.

3) Comment: "Line 138: This study include 337 cities in mainland China, but readers may not know the total number of cities in mainland China. Letting readers know the coverage of study areas in this paper would be better."

Response: Dear Professor, we really appreciate your valuable suggestion that "letting readers know the coverage of study areas in this paper would be better". According to your suggestion, we have added the details about the inclusion of cities and the coverage of study areas in our study. In fact, all 337 cities in mainland China were included in this study. A total of 333 prefecture-level administrative regions (including 293 prefecture-level cities, 30 autonomous prefectures, 7 prefectures and 3 leagues) and 4 municipalities (including Beijing, Shanghai, Tianjin, and Chongqing) were included in our analysis. Please check the descriptions in **the Study domain section, the Materials and methods part**. Thank you very much.

4) Comment: "Line 157-163: for the mosquito vector density data, the authors described the details of monitoring vector density, but missed one important information, i.e., the frequency of sampling."

Response: Dear Professor, we really agree with you that the frequency of sampling is indeed an important information. We have added the details about the frequency of sampling according to your suggestion. Please check the revision in **the Mosquito vector density data section of the Materials and methods part**. Thank you very much.

5) Comment: "Line 171-174: Daily meteorological data were obtained from 2013 to 2019, but the study period was 2013 to 2020. Why?"

Response: Dear Professor, we really thank you for giving us the opportunity to explain the issue to you here. It is correct that the study period was 2013 to 2020. In order to give the whole picture of nationwide epidemiological characteristics and transmission dynamics of dengue in China, the analyses of this study were mainly divided into two parts. Initially, we performed the descriptive analysis of local and imported cases, latitudinal gradient characteristics of cases distribution and seasonal patterns of epidemic using dengue cases from 2013 to 2020. However, the main aim of this study was to explore the dynamics of dengue transmission in China, especially in the last decade. To achieve this goal, we developed metapopulation networks coupled with meteorological and population migration data to estimate transmission parameters of dengue. We aimed to summarize the characteristics of dengue epidemics in the last 10 years prior to the outbreak of COVID-19 (the outbreak of COVID-19 began in China since early 2020), rather than the special case of the outbreak. Therefore, in this part of metapopulation network modelling, dengue cases from 2013-2019 were used and incorporated into the model. Meteorological data corresponding to the period of 2013-2019 were also used to estimate some parameters including infection rates between humans and mosquitoes and so on. This is the reason why daily meteorological data for 2013-2019 was included, while the whole study period we depicted was 2013-2020. In addition, we would like to explain to you that this study did not estimate the parameters of dengue epidemic after the outbreak

of COVID-19, mainly considering that the population migration during this period was severely affected by the outbreak, and the limited population migration data currently available during the outbreak could not accurately reflect the real situation at that time because the government adopted a very strict population movement containment strategy. If we were to integrate this untrue population migration data into our metapopulation network models, it would have an immeasurable impact on the estimates. Our further studies will explore this issue. We really hope that the above clarifications can be understood by you. Thank you very much.

6) Comment: "Line 446-447: “We found that the variation of R_{eff} for dengue fever in cities across the nation”. This sentence is confusing."

Response: Dear Professor, thank you very much for pointing out this mistake. We are very sorry for the poor language expression in the manuscript for the first round of submission. We have revised the sentence in **the 3rd paragraph of the Intrinsic dynamic characteristics of dengue virus transmission section, the Results part**. In addition, based on your comment, we have also reviewed the entire manuscript for any spelling, punctuation, or grammatical errors. Some errors have been revised. Please check them. Thank you again for your valuable comment.

7) Comment: "Line 473-475: the authors claimed that “Our study of more than 90,000 cases of dengue fever reported to the NNDSS during 2013-2020 in China gives the most comprehensive quantification of national burden and time-evolving intrinsic dynamic characteristics of the disease up to now.” Use of strong words such as “the most” should be careful."

Response: Dear Professor, we really appreciate your rigorous attitude and this insightful suggestion. According to your suggestion, we have removed the strong word "most", and rephrased the sentence in a more objective way. Please check it in **the 1st paragraph of the Discussion part**. Thank you very much.

8) Comment: "What are the limitations of this study? There should be a limitation section."

Response: Dear Professor, thank you very much for this constructive suggestion. We have discussed the limitation of our study. Please check in **the 7th paragraph of the Discussion part**. Thank you very much.

Once again, we appreciate your valuable comments.

Yours sincerely

To Reviewer 2:

Dear professor, thank you very much for your encouragement “their statistical and modeling analyses identified several interesting findings, including the geographic expansion of dengue distribution and seasonality and key drivers of dengue spread in China. Their research outputs could be useful to inform local dengue control policies”. We really thank you for the insightful and constructive comments, although you feel that some issues should be clarified to improve the manuscript. We have carefully revised the manuscript according to your suggestions. We hope the revisions will make you feel appropriate and finally be approved by you. Our responses to the comments have been arranged one by one in the following section.

1) Comment: "In supplementary lines 150 to 277, the authors had two duplicate sections of dengue diagnostic methods, with inconsistent details. Could you provide a summary table of probable, clinical diagnosed, and lab-confirmed diagnostic standards?".

Response: Dear Professor, thank you very much for this comment. We apologize for the lack of clarity in the texts of the supplementary file. Actually, these two parts are not duplicate sections. These two sections correspond to the diagnostic criteria for dengue fever before 2018 and after 2018, respectively. The diagnostic criteria for dengue fever can also be found in the literature (please refer to: *Chuanxi Li, et al. Projecting future risk of dengue related to hydrometeorological conditions in mainland China under climate change scenarios: a modelling study. The Lancet Planetary Health, 2023: e397-e406*). Based on your comment, we have revised the descriptions of the two sections of dengue diagnostic methods by adding distinct subheadings for each section. Please check them in **the Supplementary Material file**. Thank you very much.

2) Comment: "For laboratory diagnosis, many important information has been

missing. For example, the protocol of IgM, IgG dengue serology, RT-PCR, virus isolation, and NS1 antigen test. How about the case misclassification rate for laboratory, clinical and epidemiological diagnosis? Proportion of primary and post-primary dengue cases in local/imported cases? Proportion of cases with different severity (i.e., dengue fever, dengue hemorrhagic fever, dengue shock syndrome). The authors could discuss the influence of these case definition factors on the estimated results".

Response: Dear Professor, thank you very much for this comment. We are very grateful to you for giving us the opportunity to elaborate on this issue. In fact, we have provided the descriptions about dengue fever disease diagnostic criteria at **the beginning of page 6 of the Supplementary Material file**. In mainland China, the National Health Commission is responsible for establishing and regularly updating diagnostic criteria for various infectious diseases. The comprehensive diagnosis of dengue cases was made according to the epidemiological evidence, clinical manifestations and laboratory examination results of the patients. All dengue fever cases were diagnosed according to the diagnostic criteria for Dengue Fever (WS 216-2008 and WS 216-2018). The information on the diagnostic criteria can also be found in a recently published literature (please refer to: *Chuanxi Li, et al. Projecting future risk of dengue related to hydrometeorological conditions in mainland China under climate change scenarios: a modelling study. The Lancet Planetary Health, 2023: e397-e406*). The important information about laboratory diagnosis including the protocol of IgM, IgG dengue serology, RT-PCR, and virus isolation can be found in the appendix document. According to your comment, we have added the details about the case definitions in **the Dengue fever incidence data section of the Materials and methods part**. More details can be found in **the Supplementary Material file**. Please check them. Thank you very much.

For the comment about the influence of the case definition factors on the estimated results, we really agree with you that the factors (e.g., case misclassification

rate) may have an impact on the results of our study. We really appreciate your insightful comment and would like to explain the issue to you here. Since 2003, China has established a national enhanced surveillance system for infectious diseases. The National Notifiable Diseases Surveillance System (NNDSS) is such a platform for this national enhanced surveillance. The data of dengue fever cases used in our study were obtained from the NNDSS of the China CDC. It is currently the largest national surveillance platform of infectious diseases in China. Although it is currently the largest surveillance platform in the country, the information it can provide is still basic, in particular, especially the detailed information of cases (such as classification on primary and post-primary cases, and cases with different severity, etc.) is not yet available. According to your suggestion, we have discussed the issue in **the 7th paragraph of the Discussion part**. Please check it. Thank you very much.

3) Comment: "In section "Metapopulation network modelling for local dengue cases" in Main Text, the SI-SIR model might be oversimplified for modeling dengue transmission between mosquitoes and human populations. The authors could use the SEI-SEIR model framework (Ref: Climate predicts geographic and temporal variation in mosquito-borne disease dynamics on two continents. *Nat. Commun.* 12, 1233 (2021)) that addresses the mosquito life-history dynamics and dependency of virus development rate to temperature".

Response: Dear Professor, you have extensive knowledge in the field of infectious disease dynamics modeling. Thank you very much for this valuable suggestion. We really cherish this opportunity to explain the issue to you here. Actually, in general dengue virus has an incubation period of 4-8 days in the human body (please refer to: *Guzman, M. G., & Harris, E. (2015). Dengue. The Lancet, 385(9966), 453-465.*). It seems more mathematically reasonable to use a SEIR model containing the exposed compartment. However, in the setting of SIR model the information on the exposed compartment is implicitly included in the transmission process from S (the susceptible

compartment) to I (the infected compartment). In recent years, researchers tended to use mathematical models without exposed compartments to explore the spread of dengue (please refer to: [1] *Supriatna, A., et al. (2022). Age-Dependent Survival Rates in SIR-SI Dengue Transmission Model and Its Application Considering Human Vaccination and Wolbachia Infection in Mosquitoes. MATHEMATICS, 10(21)*; [2] *Tay, C. J., et al. (2022). Dengue epidemiological characteristic in Kuala Lumpur and Selangor, Malaysia. Mathematics and Computers in Simulation, 194, 489–504*; [3] *Chen, Y., et al. (2022). An ensemble forecast system for tracking dynamics of dengue outbreaks and its validation in China. PLoS Comput Biol, 18(6), e1010218*; [4] *Zeng, Q., et al. (2023). Dengue transmission dynamics prediction by combining metapopulation networks and Kalman filter algorithm. PLoS Negl Trop Dis, 17(6), e0011418*). Meanwhile, previous studies comparing the results of SIR and SEIR compartment models for inferring dengue transmission parameters showed that whether the exposed compartment was added in the model had little influence on the inference of model parameters (please refer to: *Valencia, V. A. N. (2023). Using compartmental models and Particle Swarm Optimization to assess Dengue basic reproduction number for the Republic of Panama in the 1999-2022 period. Heliyon.*). Moreover, another important reason for adopting the SI-SIR model instead of the SEI-SEIR model in our study is to consider the complexity of the metapopulation network model. We do not incorporate the exposed compartment in the inference system in order to ensure the stability of the high-dimensional model and the identifiability of the system, while satisfying the accuracy of parametric inference. You can see that our inference model is a huge system of metapopulation network considering differential equation models with 337 cities linked to each other based on population movement, meaning that it needs to solve at least one large matrix of 337×337 dimensions (please see the details about the model in **the Supplementary Materials file**). Also, we used the proposed data assimilation algorithm called the ensemble adjustment Kalman filter (EAKF) approach to make posteriori inferences

about the parameters and calibrates the model. By adopting the SI-SIR model as a basic model in the metapopulation network system, we aimed to reduce the complexity and uncertainty of the inference system as much as possible on the premise of satisfying the accuracy of parameter inference. This approach has been widely used in previous studies (please refer to: [1] *Shaman J, et al. Forecasting seasonal outbreaks of influenza. PNAS. 2012;109: 20425-20430*; [2] *Yamana TK, et al. Superensemble forecasts of dengue outbreaks. J R Soc Interface. 2016; 13: 20160410*). A recent study proposed to using a “simple” SIR model as a basic model in the metapopulation network to estimate the transmission risk of COVID-19 (please refer to: *Pei, S., et al. (2021). Burden and characteristics of COVID-19 in the United States during 2020. Nature, 598(7880), 338-341.*). Moreover, our previous studies also confirmed the effectiveness of this approach for inferring the risk of dengue transmission in our country (please refer to: [1] *Yuliang Chen, et al. An ensemble forecast system for tracking dynamics of dengue outbreaks and its validation in China. PLoS Comput Biol. 2022: 18(6)*; [2] *Zeng, Q. et al. Dengue transmission dynamics prediction by combining metapopulation networks and Kalman filter algorithm. PLoS Negl Trop Dis 2023: 17, e0011418*). We hope that the above explanation can clarify this issue and gain your understanding. Thank you again.

4) Comment: "It would be helpful if the authors could find some evidence or literature to support the assumption that the human populations remain fully susceptible to dengue infection in China".

Response: Dear Professor, thank you very much for this constructive suggestion. We really thank you for giving us this opportunity to explain the issue to you here. Dengue virus (DENV) is represented by four distinct serotypes (DENV1-4) that co-circulate in tropical and sub-tropical regions and can repeatedly infect humans (please refer to: *Nemirov K, et al. Preclinical proof of concept of a tetravalent lentiviral T-cell vaccine against dengue viruses. Front. Immunol. 2023, 14:1208041*).

Meanwhile, previous studies have shown that infection with any DENV serotype results in a short period of heterotypic immunity (that is, immunity against another serotype) (please refer to: *Guzman MG, et al. Dengue infection. Nat Rev Dis Primers. 2016;2:16055*). All four serotypes of DENV (DENV-1 to DENV-4) are commonly transmitted across different regions in mainland China, showing significant spatial heterogeneity (please refer to: *Tiantian Wu, et al. Dengue fever and dengue virus in the People's Republic of China, Rev Med Virol. 2022;32:e2245*). It is because of the reality of the prevalence of these different serotypes of DENV in China that we have made corresponding assumptions about the model. According to your suggestion, we have added some evidence to support the assumption. Please check the revisions in **the Metapopulation network modelling for local dengue cases section of the Materials and methods part, and the Dengue fever transmission model section in Supplementary Material file**. Thank you again for the important suggestion.

5) Comment: "In Table 1, the average of daily effective reproduction number is estimated to be larger than 1 in each considered city and province. This may suggest that dengue has been endemic in these cities in 2019? However, the average force of infection is estimated to be very low, which might be inconsistent with the estimates of R_{eff} . What's the average generations of infected people caused by each imported / local case?"

Response: Dear Professor, thank you very much for this insightful comment. We really cherish this opportunity to explain the issue to you here. First, please allow us to explain the meanings of the indicators including the effective reproduction number (R_{eff}) and the force of infection a little more for better understanding. Unlike other infectious diseases that can be transmitted directly from person to person (e.g., influenza), dengue fever is a vector-borne infectious disease and is transmitted only by mosquito bites. Thus, the R_{eff} indicates the average number of infected persons infected by a diseased infected individual during the period of infection, depending on

the probability of an infected person infecting a susceptible mosquito becoming infected and the probability of an infected mosquito infecting a susceptible person. With regard to your comment that “what’s the average generations of infected people caused by each imported/local case”, the R_{eff} indicator actually provides the information on the average generations of infected people caused by each imported/local case. In our study, we initially estimated the daily fluctuating R_{eff} for individual cities and then the average values were calculated for the main cities experiencing significant seasonal dengue epidemics in September and October (please see Table 1). Your comment is correct. In general, the indicator R_{eff} exceeding 1 indicates an epidemic of the disease occurring in the region. As illustrated in **Figure 5**, the analysis results demonstrate that the effective reproduction number R_{eff} across cities is commonly greater than 1 in several specific months from July to November, suggesting that the current epidemic of dengue in China is not a long-term sustained epidemic state. The results of the size and timing of seasonal dengue outbreaks based on model inference further confirm the seasonal fluctuations of dengue epidemic, but also suggest that seasonal dengue transmission is spatially heterogeneous. Corresponding descriptions have been added in **the Abstract and Results parts** to make it clearer. In addition, we would like to point out that, the results of the R_{eff} estimated in our study is reasonable and can be supported by some existing studies about the R_{eff} estimation of dengue fever in some areas of China (please refer to: *Liu, Y., Lillepold, K., Semenza, J. C., Tozan, Y., Quam, M. B. M., & Rocklöv, J. (2020). Reviewing estimates of the basic reproduction number for dengue, Zika and chikungunya across global climate zones. Environmental Research, 182, 109114.*

As we have described above, dengue fever is a vector-borne infectious disease and is transmitted only by mosquito bites. Thus, it corresponds to the force of infection that an infected mosquito (rather than an infected person) brings directly to a susceptible population. Further, we can understand it this way, the indicator of force of infection indicates the probability of a susceptible person becoming infected with

dengue, depending on the probability that the susceptible person acquires the DENV from an infected mosquito. On the basis, we can know that, the meanings of the two indicators are different, so there are differences in their estimates. According to your valuable comment, we have added detailed meanings of the R_{eff} and force of infection in this revised version of manuscript for better understanding. Please check the revisions in **the Transmission dynamics and parameters of dengue fever section, the Materials and methods part**. Thank you again for this constructive comment.

6) Comment: "In supplementary Fig. 30, the authors analyzed each year from 2013 to 2018. However, the main text seems mainly taking about the results for 2019. The authors may wish to discuss the interannual variations and corresponding drivers."

Response: Dear Professor, thank you very much for this important suggestion. According to your suggestion, we have compared the inferred results of the interannual variations in this study from 2013 to 2019. The descriptions have been added in **the 5th paragraph of the Intrinsic dynamic characteristics of dengue virus transmission section, the Results part**. In addition, we have also discussed the interannual variations and potential drivers based on your suggestion. Please check it in **the 3rd paragraph of the Discussion part**. Thank you again.

7) Comment: "Refs. [1] Climate predicts geographic and temporal variation in mosquito-borne disease dynamics on two continents. Nat. Commun. 12, 1233 (2021)."

Response: Dear Professor, thank you for recommending the above literature, which we have read and found helpful in illustrating the research points. Therefore, we have cited it in **the 3rd paragraph of the Discussion part**. Thank you very much.

Once again, we appreciate your valuable comments.

Yours sincerely

To Reviewer 3:

Dear professor, thank you very much for your encouragement “this is an overall welcome manuscript describing the burden of dengue in China”. We really thank you for the insightful and constructive comments. We have carefully revised the manuscript according to your suggestions. We hope the revisions will make you feel appropriate and finally be approved by you. Our responses to the comments have been arranged one by one in the following section.

1) Comment: “Abstract: Avoid terms such as "super outbreaks". No outbreak in China has been as large as the one we are currently seeing in Brazil, with 4 million cases within a matter of January till April 2024, or outbreaks such as seen in SE Asia over the past decades. “Super” is therefore misleading, especially as the next sentence says 95,000 cases. That is truly a small number compared to the 100 million cases worldwide and compared to the large population size of China.”

Response: Dear Professor, thank you very much. We really appreciate your insightful suggestion and your rigorous academic attitude. Your comment that "no outbreak in China has been as large as the one we are currently seeing in Brazil, That is truly a small number compared to the 100 million cases worldwide and compared to the large population size of China" is very correct. We have revised inappropriate expression in **the Abstract part** according to your suggestion. Other parts of the text using the word "super" have also been revised accordingly. Please check them. Thank you.

2) Comment: “Abstract: What do you mean by your statement “it suggested that dengue did not seem to become an endemic disease at present”. Do you mean it is an epidemic disease? It is already endemic!”

Response: Dear Professor, thank you very much. We really appreciate your insightful suggestion and your rigorous academic attitude. We feel sorry for this mistake and the poor language expressions in the manuscript of the first round of submission. Actually,

we would like to state that the effective reproduction number R_{eff} across cities is commonly greater than 1 in several specific months from July to November, suggesting that the current epidemic of dengue in China is not a long-term sustained epidemic state. The results of the size and timing of seasonal dengue outbreaks based on model inference further confirm the seasonal fluctuations of dengue epidemic, but also suggest that seasonal dengue transmission is spatially heterogeneous. According to your comment, we have stated it more clearly. Please check the revision in **the Abstract part**. Thank you again for the important suggestion.

3) Comment: “Introduction: How can the authors call 40,000 cases a “super outbreak”? The authors also fail to state that urbanization and increasing population densities are the main drivers for dengue, not climate change. Obviously, climate and weather parameters are important for the vector, and indeed the vector can expand to naïve areas with climate change. However, in areas where the ecological conditions are already favorable to the vector, the increasing incidence of dengue is due to increasing population densities.”

Response: Dear Professor, we really appreciate your comment. According to your suggestion, we have revised inappropriate expression “super outbreak”. Please check it in **the 2nd paragraph of the Introduction part**. In addition, we really agree with you that “the authors also fail to state that urbanization and increasing population densities are the main drivers for dengue, not climate change”. According to your suggestion, we have described the effects of urbanization and population growth as the main drivers on dengue epidemics. Please check the revisions in **the 2nd paragraph of the Introduction part**. Thank you.

4) Comment: “The authors would need to check whether increasing population size in the Southern Cities are correlated with the increasing incidence”.

Response: Dear Professor, we really appreciate you giving us this valuable comment.

We really cherish this opportunity to explain the issue to you here. In fact, urbanization, population growth and human mobility are the main drivers on dengue epidemics (please refer to: *Messina JP, et al. The current and future global distribution and population at risk of dengue. Nat Microbiol. 2019; 4(9):1508-1515*). You have also pointed out this constructive comment above. Whether the increasing incidence of dengue fever in southern Chinese cities is related to the increase in population size? We speculate that there is a close relationship between them. That's because previous studies from southern Chinese cities have shown that high population density and frequent population movements are related to high incidence of dengue, and population density and movements are main drivers shaping dengue's distribution and dynamics (please refer to: [1] *Zhang Z, et al. The increasing menace of dengue in Guangzhou, 2001-2016: the most important epicenter in mainland China. BMC Infect Dis. 2019;19(1):1002*; [2] *Zeng, Q. et al. Dengue transmission dynamics prediction by combining metapopulation networks and Kalman filter algorithm. PLoS Negl Trop Dis 2023; 17, e0011418*; [3] *Keke Liu, et la. The driver of dengue fever incidence in two high-risk areas of China: A comparative study. Scientific Reports 2019; 9, 19510*). The issues have been discussed in our manuscript based on your comment. However, in order to investigate whether increasing population size in the Southern Cities are correlated with the increasing incidence, from the perspective of methodological requirements, dengue surveillance data over a longer period of time (often spanning decades) is necessary. For example, a previous study in Singapore used nearly 40 years of dengue surveillance data to explore whether an increase in dengue incidence is related to population growth (please refer to: *Struchiner CJ, et al. Increasing Dengue Incidence in Singapore over the Past 40 Years: Population Growth, Climate and Mobility. PLoS ONE. 2015; 10(8): e0136286*). To be honest, this is beyond the scope of this present study. In fact, the main aim of our study was to give the whole picture of the epidemiological characteristics underpinning the outbreaks of DENV in China, in particular quantifying time-evolving parameters of

dengue spread dynamics and burden associated with epidemics waves. We will systematically explore the relationship between dengue incidence and urbanization as well as population growth in China in future studies. We have clarified this issue and hope to gain your understanding. Thank you again.

5) Comment: “References: Several references are old (eg Ref 1, 2, 6 ,7 etc) and should be updated. There are also some other references related to China that the authors could add. Here some suggested references:

[1] Global burden for dengue and the evolving pattern in the past 30 years. Yang X, Quam MBM, Zhang T, Sang S. *J Travel Med.* 2021 Dec 29;28(8):taab146. doi: 10.1093/jtm/taab146. PMID: 34510205

[2] Discrepancies in dengue burden estimates: A comparative analysis of reported cases and Global Burden of Disease Study, 2010-2019. Lee SY, Shih HI, Lo WC, Lu TH, Chien YW. *J Travel Med.* 2024 May 2:taae069. doi: 10.1093/jtm/taae069. Online ahead of print. PMID: 38696416

[3] Reviewing estimates of the basic reproduction number for dengue, Zika and chikungunya across global climate zones. Liu Y, Lillepold K, Semenza JC, Tozan Y, Quam MBM, Rocklöv J. *Environ Res.* 2020 Mar;182:109114. doi: 10.1016/j.envres.2020.109114. Epub 2020 Jan 3. PMID: 31927301

[4] Global burden and trends of neglected tropical diseases from 1990 to 2019. Lin Y, Fang K, Zheng Y, Wang HL, Wu J. *J Travel Med.* 2022 May 31;29(3):taac031. doi: 10.1093/jtm/taac031. PMID: 35238925

[5] The Lancet Commission on dengue and other Aedes-transmitted viral diseases. Wilder-Smith A, Lindsay SW, Scott TW, Ooi EE, Gubler DJ, Das P. *Lancet.* 2020 Jun 20;395(10241):1890-1891. doi: 10.1016/S0140-6736(20)31375-1.

[6] Increasing Dengue Incidence in Singapore over the Past 40 Years: Population Growth, Climate and Mobility. Struchiner CJ, Rocklöv J, Wilder-Smith A, Massad E. *PLoS One.* 2015 Aug 31;10(8):e0136286. doi: 10.1371/journal.pone.0136286.

eCollection 2015. PMID: 26322517”

Response: Dear Professor, we really agree with your suggestion. Accordingly, we have updated the references in **the Introduction part**. In addition, we have also added the references you provided above. Please check them in **the References part**. Thank you very much.

Once again, we appreciate your valuable comments

Yours sincerely

Reviewers' Comments:

Reviewer #2:

Remarks to the Author:

The authors have addressed previous comments. My pleasure to recommend the acceptance of this great work.

Reviewer #3:

Remarks to the Author:

Thank you for addressing the comments and suggestions.